# Targeting bivalency de-represses Indian Hedgehog and inhibits self-renewal of colorectal cancer-initiating cells

Evelyne Lima-Fernandes [1,2], Alex Murison[2], Tiago da Silva Medina[2], Yadong Wang[2], Anqi Ma[3], Cherry Leung[2], Genna M. Luciani [1,2], Jennifer Haynes[2], Aaron Pollett[4,5], Constanze Zeller[2], Shili Duan[6], Antonija Kreso[2], Dalia Barsyte-Lovejoy [1], Bradly G. Wouters[2,6,7], Jian Jin [3], Daniel D. De Carvalho [2,6], Mathieu Lupien [2,6,8], Cheryl H. Arrowsmith[1,2,6] & Catherine A. O'Brien [2,4,6,9,10]

In embryonic stem cells, promoters of key lineage-specific differentiation genes are found in a bivalent state, having both activating H3K4me3 and repressive H3K27me3 histone marks, making them poised for transcription upon loss of H3K27me3. Whether cancer-initiating cells (C-ICs) have similar epigenetic mechanisms that prevent lineage commitment is unknown. Here we show that colorectal C-ICs (CC-ICs) are maintained in a stem-like state through a bivalent epigenetic mechanism. Disruption of the bivalent state through inhibition of the H3K27 methyltransferase EZH2, resulted in decreased self-renewal of patient-derived C-ICs. Epigenomic analyses revealed that the promoter of Indian Hedgehog (*IHH*), a canonical driver of normal colonocyte differentiation, exists in a bivalent chromatin state. Inhibition of EZH2 resulted in de-repression of IHH, decreased self-renewal, and increased sensitivity to chemotherapy in vivo. Our results reveal an epigenetic block to differentiation in CC-ICs and demonstrate the potential for epigenetic differentiation therapy of a solid tumour through EZH2 inhibition.

[1] Structural Genomics Consortium, University of Toronto, Toronto, ON M5G1L7, Canada. [2] Princess Margaret Cancer Centre, University Health Network, Toronto, ON M5G1L7, Canada. [3] Mount Sinai Center for Therapeutics Discovery, Departments of Pharmacological Sciences and Oncological Sciences, Tisch Cancer Institute, Icahn School of Medicine at Mount Sinai, New York, NY 10029, USA. [4] Department of Laboratory Medicine and Pathobiology, University of Toronto, Toronto, ON M5S1A8, Canada. [5] Lunenfeld-Tanenbaum Research Institute Toronto, Toronto, ON M5G1X5, Canada. [6] Department of Medical Biophysics, University of Toronto, Toronto, ON M5G1L7, Canada. [7] Department of Radiation Oncology, University of Toronto, Toronto, ON M5G1L7, Canada. [8] Ontario Institute for Cancer Research, Toronto, ON M5G1L7, Canada. [9] Department of Physiology, University of Toronto, Toronto, ON M5G1L7, Canada. [10] Department of Surgery, Toronto General Hospital, Toronto, ON M5G2C4, Canada. Correspondence and requests for materials should be addressed to C.H.A. (email: carrow@uhnresearch.ca) or to C.A.O'. (email: Catherine.OBrien@uhnresearch.ca)

The cancer stem cell model posits that a subset of tumour cells, cancer stem cells or cancer-initiating cells (C-IC), are endowed with the capacity for self-renewal, which is characterised by the ability to initiate and sustain tumour growth in xenotransplantation assays[1]. C-ICs have been described in multiple cancer types, including colorectal cancer (CRC), and are often associated with chemotherapy resistance and disease recurrence[2,3]. In CRC, an intestinal stem cell signature of the tumour correlated with decreased patient survival and predicted relapse[4], further stressing the clinical relevance of stemness in this disease. The original model of C-ICs as static entities at the apex of a cellular hierarchy has been challenged by evidence of phenotypic plasticity involving interconversion from non-C-IC to C-IC states in response to both intrinsic and extrinsic stimuli[5]. Such reprogramming of cellular identity within a tumour cell population raises the question of whether C-ICs retain the capacity to terminally differentiate into non-C-ICs. Thus, further work is required to understand the molecular mechanisms that drive C-ICs to irreversibly exit the stem-like state, and whether differentiation therapy represents a clinical opportunity to target solid tumour C-ICs.

Recent evidence indicates that epigenetic regulation of transcriptional programmes is a key driver of self-renewal capacity, the defining feature of the C-IC state. In the context of CRC, epigenetics have been shown to contribute to the C-IC state by modulating key pathways such as Wnt signalling, where high activity has been shown to designate the colorectal cancer-initiating cell (CC-IC) population[5]. One of the initial examples of an epigenetic regulator influencing CRC growth was LSD1, an enzyme catalyzing the demethylation of H3K4me1/2 and H3K9, which contributes to CC-IC self-renewal through downregulation of the Wnt pathway antagonist DKK1[6]. Another example is Bmi1, a subunit of the PRC1 complex, which binds to H3K27me3 to repress transcription and has been well documented for its role in tumorigenicity. Bmi1 inhibition reduced self-renewal of CC-ICs by decreasing Wnt pathway activity through reduction of β-catenin levels[7]. These initial studies point to the importance of epigenetic regulators in maintaining CC-IC self-renewal through their contribution to Wnt pathway activation. However, a key question that remains is whether CC-ICs can be reprogrammed through targeting epigenetic modifiers to alter cell fate specification and drive terminal differentiation.

To better understand how epigenetic regulation defines and maintains the CC-IC state, we used a collection of epigenetic chemical probes[8] to interrogate a panel of patient-derived CRC models. We identified Enhancer of Zeste Homologue 2 (EZH2), as a key contributor to the CC-IC state. EZH2 is the catalytic subunit of Polycomb Repressive Complex 2 (PRC2) which tri-methylates H3K27, and is upregulated in multiple cancer types including CRC, where its expression level correlates with worse prognosis[9,10]. EZH2 expression levels have also been shown to be higher in C-ICs from a broad range of tumours, including breast, ovarian and leukaemia[11].

EZH2 has an established role in maintaining embryonic stem cells (ESCs) in an undifferentiated state through repression of lineage commitment and differentiation genes[12,13]. In ESCs, the promoters of genes involved in differentiation are often found in a bivalent state, defined by the simultaneous presence of both the activating histone mark H3K4me3 and the repressive histone mark H3K27me3. The presence of both marks results in silencing of developmental programmes and maintenance of pluripotency[14]. These bivalent genes are thought to be poised for expression upon loss of repressive H3K27me3, which triggers transcription of differentiation programmes[12–14]. Therefore, as the catalytic subunit of PRC2, EZH2 plays a key role in maintaining bivalency at lineage commitment genes in ESCs.

Initial work in cancer suggests that bivalency may be playing a similar role in the context of cancers with activating mutations in EZH2, such as germinal centre B cell lymphomas, where H3K27me3-mediated repression of bivalent promoters results in suppression of differentiation[15]. In contrast, the role of bivalency in tumours that overexpress EZH2 but lack activating mutations has not been established. Instead, a number of other bivalency-independent mechanisms have been identified in a tissue-specific manner. In the context of CRC, previously published work using commercially available CRC cell lines showed that EZH2 inhibition resulted in downregulation of the Wnt/β–catenin pathway and decreased CC-IC tumour-initiating capacity and mammosphere formation[16]. Despite extensive research on EZH2 in cancer, there is currently no evidence showing that similar to ESCs, EZH2 represses differentiation programmes in C-ICs through maintenance of bivalency. If bivalency is maintaining the C-IC state through repression of differentiation programmes, it will be important to determine whether, similar to ESCs, this state is reversible upon loss of H3K27me3. Furthermore, a better understanding of the role of bivalency in the context of cancer could lead to novel epigenetic therapeutic strategies to promote C-ICs to terminally differentiate.

Here, we identify a role for EZH2 in maintaining the CC-IC state, in part, through repression of the bivalent promoter of Indian Hedgehog (*IHH*), a member of the Hedgehog pathway and a canonical marker of colonocyte differentiation in normal intestinal development. Our results demonstrate that, similar to ESCs, CC-ICs depend on bivalency to maintain self-renewal through transcriptional repression of a lineage-specific differentiation gene, and that this state can be targeted through EZH2 inhibition.

## Results

**Targeting EZH2 reduces growth of patient-derived CC-ICs.** Utilising a previously established serum-free, growth factor-enriched spheroid culture that enriches for CC-ICs in patient-derived CRC samples[17], we screened a collection of 22 selective chemical probes that inhibit epigenetic regulatory proteins using viable cell count as a readout (Supplementary Figure 1a). Each probe was used at a single concentration aiming for an estimated cellular IC$_{90}$ concentration[18] (Fig. 1a, Supplementary Table 1). Three chemical probes reduced the growth of a patient-derived CC-IC enriched culture, POP92, by more than 50% (Fig. 1a). These growth inhibitory probes included the BET Bromodomain BRD2/3/4 inhibitor JQ1, an antagonist of the Bromodomains of BRPF1/2/3, and the EZH1/2 inhibitor UNC1999[19]. The epigenetic probes screen was repeated in a broader range of CRC samples grown as patient-derived organoids (PDO), a system recently shown to maintain the heterogeneity of primary patient samples[20] (Supplementary Table 2). We consistently observed strong growth suppression, with both BET and EZH2 inhibitors (Fig. 1b). Given the important role of EZH2 in maintaining self-renewal of ESCs through silencing of differentiation programmes, we focused on investigating EZH2 as a potential driver of CC-IC self-renewal and tumour growth. We next confirmed a dose-dependent reduction in growth of the PDOs (Supplementary Figure 1b-c) and CC-IC enriched cultures (Fig. 1c) upon treatment with UNC1999, whereas the chemically similar, but inactive compound UNC2400[19] showed no effect. The decrease in cell growth following UNC1999 was concomitant with the decrease in cellular levels of H3K27me3 and H3K27me2, with an IC$_{90}$ of 3 μM (Fig. 1d and Supplementary Figure 1d).

Additional EZH2 inhibitors such as GSK343 and GSK126 are available and have greater selectivity for EZH2 over EZH1[21]. Treatment with GSK343 and GSK126 had a similar reduction in viable cell count compared with UNC1999 across three CC-IC

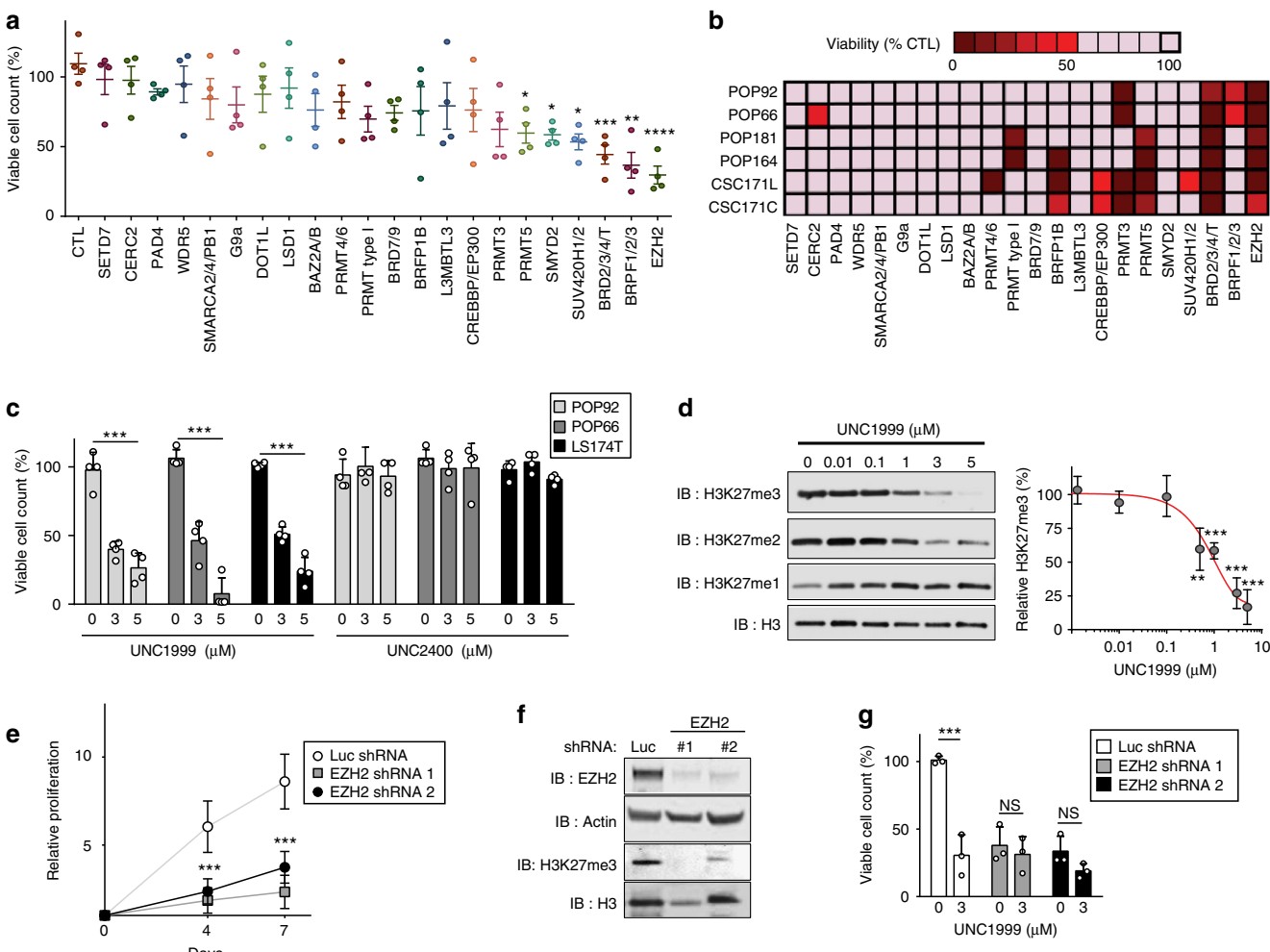

**Fig. 1** EZH2 inhibition suppresses growth in patient-derived 3D colon cancer models. **a** Viable cell count of a patient-derived spheroidal culture, POP92, treated with chemical probes for the indicated epigenetic targets for 7 days. Data shown are mean calculated as percentage of DMSO control ($n = 4$, ± SEM, one-way ANOVA). **b** Epigenetic screen performed on six patient-derived organoid models. Data are represented as percentage of DMSO control. **c** Two patient-derived (POP92 and POP66) and one commercial cell line (LS174T) grown in spheroid CC-IC enriching cultures were treated with UNC1999 (left) and its negative control UNC2400 (right) for 7 days, and viable cell count was measured and normalised to control. Data shown are mean ($n = 4$, ± SD, two-way ANOVA). **d** Representative western blot showing the decrease of H3K27me3 and H3K27me2 levels in POP92 after 7 days of treatment with increasing concentrations of UNC1999. Quantification represents the percentage of H3K27me3 levels relative to those of total H3 determined and compared to DMSO control, ($n = 4$, ± SD, Student's $t$ test). **e** Proliferation of POP92 spheroids after EZH2 knockdown using two different shRNAs, or Luciferase shRNA control, monitored by Alamarblue. Data shown are mean ($n = 4$, ± SD, two-tailed $t$ test). **f** Representative western blot showing the knockdown of EZH2 in the samples assessed for proliferation in (**e**). **g** EZH2 knockdown cells or shRNA luciferase control cells were treated with UNC1999 to show no further reduction in cell growth using viable cell count as readout. Data shown are mean ($n = 3$, ± SD, two-way ANOVA). *$P < 0.05$, **$P < 0.01$, ***$P < 0.001$, ****$P < 0.0001$. Source data are provided as a Source Data file

enriched models, supporting the role of EZH2, in maintaining growth of CC-ICs (Supplementary Figure 1e). Furthermore, genetic knockdown of EZH2 using two different shRNAs also reduced the growth of CC-IC enriched cultures (Fig. 1e–f), in line with the results obtained with the EZH2 inhibitors. Importantly, following EZH2 knockdown CC-IC enriched cultures showed no further reduction in the number of viable cells in the presence of UNC1999 (Fig. 1g). Taken together, this confirms that the growth inhibitory effect of UNC1999 is a consequence of EZH2 inhibition.

**Targeting EZH2 reduces CC-IC growth and self-renewal.** Previous reports have shown that EZH2 is highly expressed in cancer compared with normal tissue[22,23]. To assess EZH2 levels in CRC, we analysed EZH2 and H3K27me3 immunohistochemistry

staining from a tumour microarray of 283 patient samples. Staining for both EZH2 and H3K27me3 were significantly higher in CRC compared with normal intestinal tissue (Fig. 2a, b). The self-renewal properties of C-ICs have been shown to contribute to disease recurrence[2,3]. We therefore investigated the percent recurrence in patients within top and bottom quintiles of EZH2 expression. We observe a significantly greater proportion of disease recurrence in patients with tumours expressing high levels of EZH2 (31%) compared with the tumours exhibiting low levels of EZH2 expression (14%) (Fig. 2c). Using TCGA data and a published gene expression signature for colon crypt stem cells and differentiated colonocytes[24], we found that EZH2 mRNA levels in CRC samples are positively correlated with a large subset of genes, specifically expressed in the colon crypt stem cells (Fig. 2d). In contrast, EZH2 expression is inversely correlated with the majority of genes, specifically expressed in the top of the crypt,

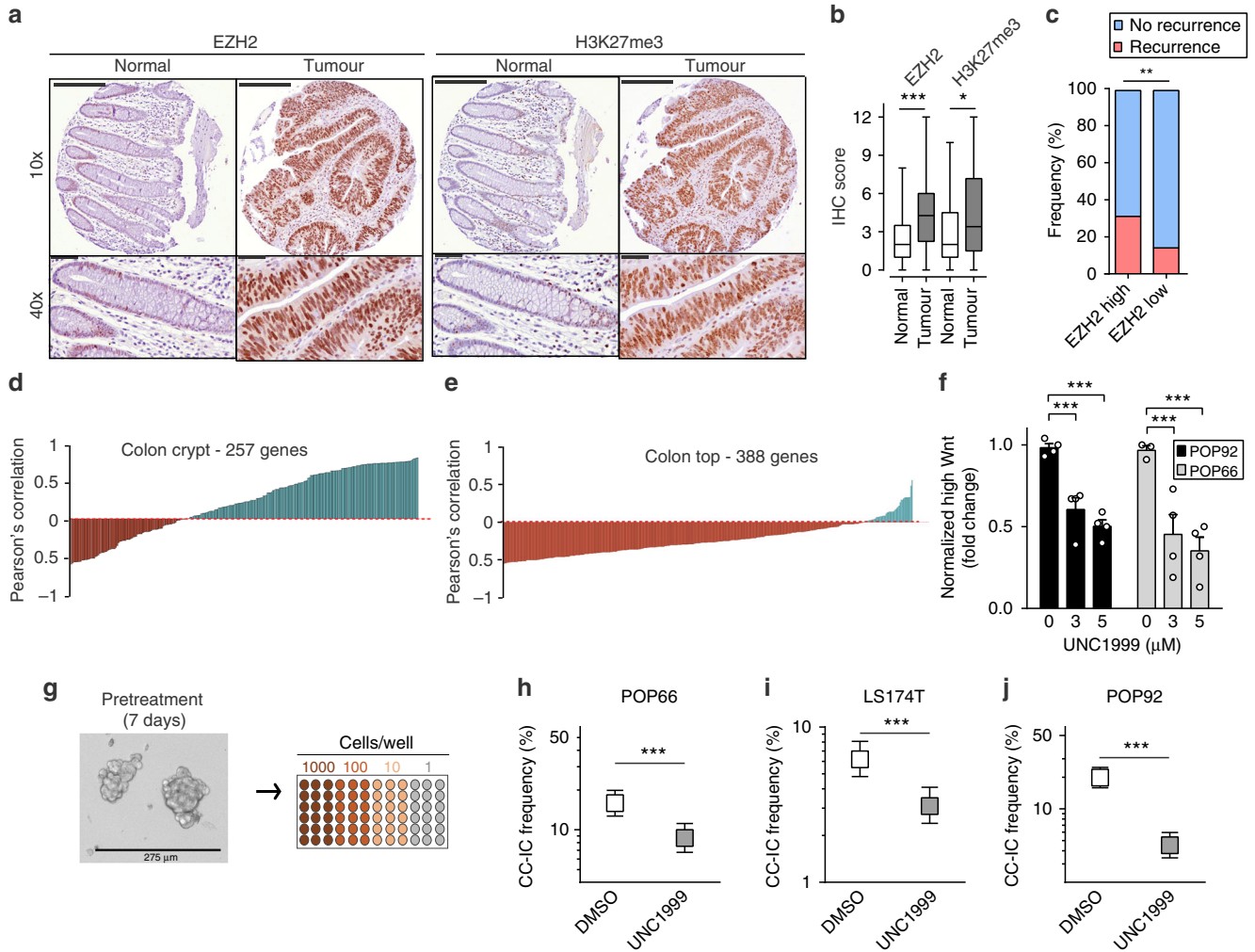

**Fig. 2** Inhibition of EZH2 reduces CC-IC frequency in vitro. **a** Tissue microarray samples processed for immunohistochemistry staining of EZH2 and H3K27me3. Pictures shown are representative of 32 patients (normal) and 251 patients (tumour). Scale bars are 200 μm (10 ×) and 50 μm (40 ×). **b** Intensity of IHC staining performed in (**a**) was scored manually as described in the Methods section, and plotted as median ± min and max, n = 32 (normal), n = 251 (tumour), Student's t test. **c** Frequency of disease recurrence in patients within the top and bottom quintiles of EZH2 IHC expression as determined in (**a**, **b**) (EZH2-low IHC score < 2.4; EZH2-High IHC score > 9.6) (Chi-square test). **d**, **e** Pearson's correlation value for EZH2 mRNA levels from TCGA CRC patient samples (COAD), compared with those of 257 genes from a colon crypt (stem cell) gene expression signature (**d**), and 388 genes from a colon top (differentiated) expression signature (**e**). Inverse correlations are plotted in red and positive correlations in blue. **f** TCF/LEF GFP Wnt reporter activity in POP92 and POP66 after 7 days of UNC1999 or UNC2400 treatment. Data are plotted normalised to DMSO control to the 10% brightest Wnt-High population (n = 4, ± SEM, two-way ANOVA). **g–j** Limiting dilution assay performed in vitro on UNC1999-pretreated cells at 3 μM for 7 days. Data shown are n = 4 (POP66, (**h**)), n = 6 (LS174T, (**i**)), n = 6 (POP92, (**j**)). Data are shown as mean  ± 95% confidence interval, frequency and probability were computed using ELDA software) *P < 0.05, **P < 0.01, ***P < 0.001. Source data are provided as a Source Data file

which represents the differentiated colonocyte compartment (Fig. 2e)[24]. Altogether, these data indicate that EZH2 expression correlates with the colonic stem cell compartment and a higher incidence of disease recurrence.

To better understand how UNC1999 treatment affects growth of CC-ICs, we performed cell cycle and apoptosis analyses on UNC1999-treated POP92 spheres (Supplementary Figure 2a–c). We observed a 9.2% increase in G1 and a 33–38% decrease in the S phase, but no significant change in apoptosis or necrosis. We further investigated whether the growth arrest in the G1/S phase following UNC1999 treatment might reflect an exit from the CC-IC state. To this end, we tested UNC1999 on two patient-derived spheroid models stably expressing a TCF/LEF GFP reporter of Wnt activity, a well-established marker of CC-ICs[5]. UNC1999 treatment resulted in a decrease of the Wnt-high expressing cells, indicating a reduction in the CC-IC fraction (Fig. 2f,

Supplementary Figure 2d–h). The reduction in Wnt reporter activity was also observed with GSK343, another EZH2 inhibitor (Supplementary Figure 2f–g). Finally, to determine whether transient EZH2 inhibition resulted in an irreversible functional decrease in CC-ICs, in vitro limiting dilution assays (LDAs) were carried out for three CC-IC enriched samples. The cells were pretreated with UNC1999 for 7 days, after which the inhibitor was removed and viable cells were seeded at limiting dose in serum-free medium lacking the inhibitor (Fig. 2g). Treatment with UNC1999 resulted in a two- to fourfold decrease in the frequency of sphere-initiating cells (Fig. 2h–j). Thus, transient inhibition of EZH2 results in an irreversible reduction in CC-ICs.

**UNC1999 reduces tumour growth and self-renewal in vivo.** To determine whether EZH2 inhibition in vivo affects tumour growth, we treated two CRC patient-derived xenograft models

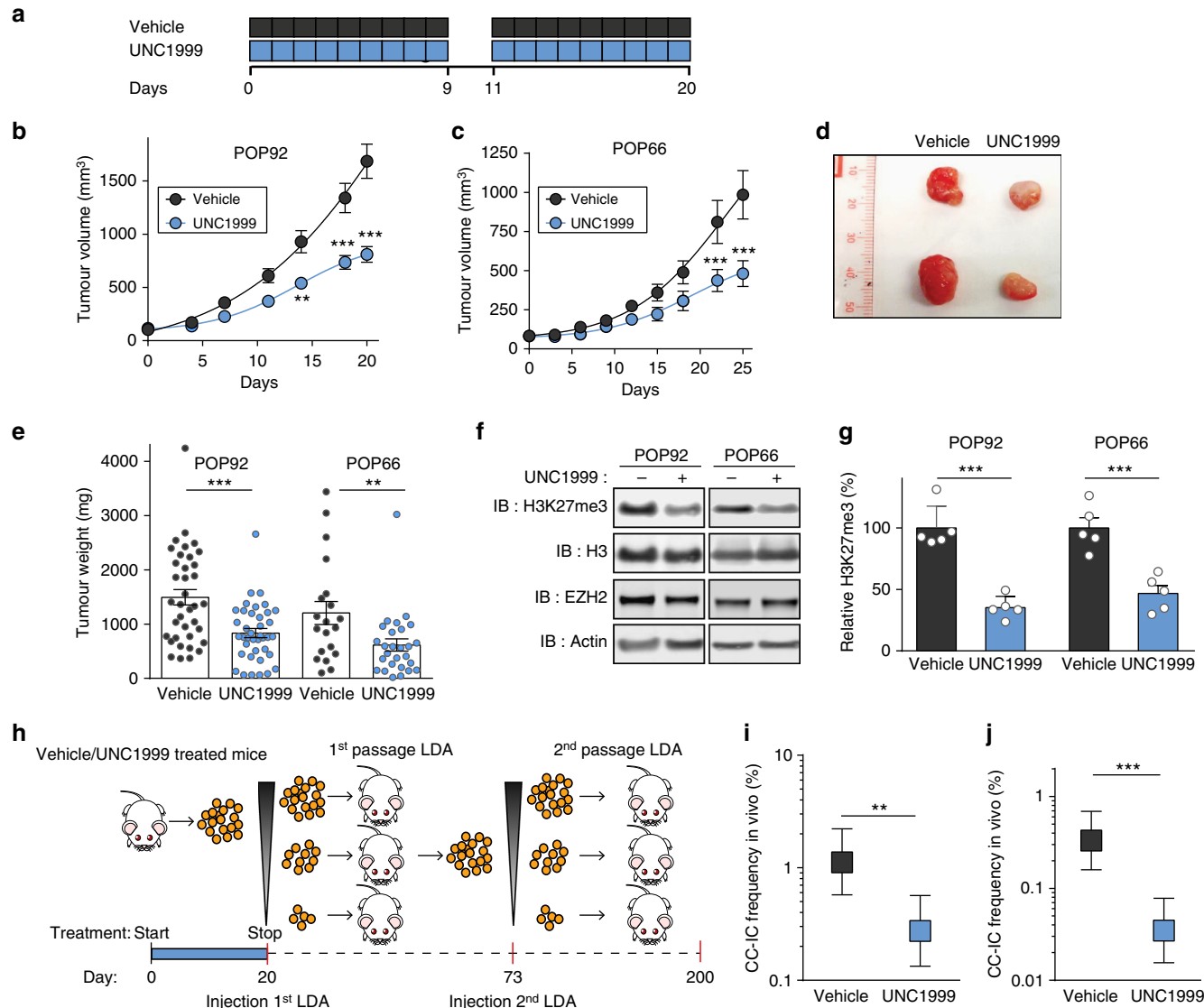

**Fig. 3** Inhibition of EZH2 reduces tumour growth and CC-IC frequency in vivo. **a** Dosing schedule for UNC1999 in vivo. Mice are dosed for 11-day cycles of 9 continuous days followed by a 2-day break. **b**, **c** Tumour volume measurements of POP92 and POP66. Mice were treated with vehicle or UNC1999 at 300 mg per kg of body weight for 20 days (**b**) or 25 days (**c**). Data shown are $n = 37$ tumours (**b**, vehicle), $n = 42$ (**b**, UNC1999), $n = 20$ (**c**, vehicle) or $n = 30$ (**c**, UNC1999) ± SEM, with two-way ANOVA. **d** Images of tumours from POP92 collected in (**b**). **e** Tumour weight measurement from xenografts collected in (**b**, **c**). Data are mean of $n = 37$ tumours (POP92 vehicle), $n = 42$ (POP92 UNC1999), $n = 20$ (POP66 vehicle) or $n = 30$ (POP66 UNC1999), ± SEM, Student's $t$ test. **f** Representative western blots showing the reduction of H3K27me3 in the xenografts collected from POP92 and POP66 samples in (**b**, **c**) respectively. **g** Quantification of the western blots in (**f**) for H3K27me3 signal over total H3 normalised to the vehicle control. Data shown are $n = 5$ tumours ± SEM with two-tailed Student's $t$ test. **h** Schematic of the serial in vivo limiting dilution assay performed in (**i**, **j**). The first passage LDA was performed using POP92 tumours collected in (**b**). Mice were treated for 20 days with UNC1999 at 300 mg/kg, tumours were collected, prepared as single cells and serially diluted and reinjected into primary recipient mice without further treatments (**i**). The second passage LDA (**j**) was performed upon growth of tumours in (**i**): samples were collected, prepared as single cells and reinjected into secondary recipient mice without further treatments (**j**). Data in (**i**, **j**) are shown as mean ± 95% confidence interval, frequency and probability were computed using ELDA software. *$P < 0.05$, **$P < 0.01$, ***$P < 0.001$. Source data are provided as a Source Data file

with vehicle control or UNC1999 at 300 mg/kg for 11-day cycles of 9 consecutive days of treatment followed by 2 days of break, which was well tolerated by the mice (Fig. 3a and Supplementary Figure 3a–b). We observed a significant reduction in both tumour volume and tumour weight in both models during the time course of treatment (Fig. 3b–e). This effect was concomitant with a decrease in H3K27me3 (Fig. 3f, g and Supplementary Figure 3c). Histological analysis revealed a modest increase in necrosis in UNC1999-treated xenografts as well as a decrease in the proliferation marker Ki67 (Supplementary Figure 3c–d). To

specifically assess the effect of EZH2 on the CC-IC population, we performed the gold standard assay for enumerating C-ICs, two serial passage in vivo LDAs on POP92 xenografts (Fig. 3h–j). Using the UNC1999-treated xenografts from the experiment shown in Fig. 3b, we serially transplanted cells in limiting dilution into primary recipient mice (1st passage LDA), and the resultant tumours subsequently into secondary recipient mice (2nd passage LDA) (Fig. 3h). The secondary transplantation was performed 52 days after the last exposure to UNC1999. These serial passage LDAs showed a significant fourfold reduction in tumour-initiating

frequency upon primary passage (Fig. 3i) and an even stronger ninefold reduction in tumour-initiating frequency upon secondary passage (Fig. 3j). Collectively, our results show that EZH2 inhibition targets CC-IC self-renewal, resulting in CC-IC exhaustion as shown by the reduction in tumour initiation in the in vivo LDA.

**The differentiation gene *IHH* is bivalently marked in CC-ICs.** To gain mechanistic insights into how EZH2 inhibition affects CC-ICs, we performed RNA-seq to compare control and UNC1999-treated POP92 CC-IC enriched cultures. A total of 50 genes were significantly downregulated as a consequence of EZH2 inhibition (Fig. 4a), including many cell cycle regulators, consistent with the observed phenotype of growth suppression (Supplementary Figure 4a). A larger group of 333 genes showed significantly increased expression after UNC1999 treatment including genes associated with gene ontology (GO) terms for metabolism, stress response and innate immune response (Supplementary Data 1, Supplementary Figure 4b). These data are consistent with previously reported cancer-specific upregulation of the Type III interferon pathway[25], and reduced expression of cell cycle genes[26] upon EZH2 inhibition. Interestingly, Gene Set Enrichment Analysis (GSEA) uncovered a significantly decreased enrichment for the Colon Crypt signature in UNC1999-treated cells (Fig. 4b). We also observed that CDX2, an intestinal differentiation marker[27], was upregulated following UNC1999 (Supplementary Data 1). Taken together, these data suggest that UNC1999 might induce exit of the stem-like state by upregulating a differentiation programme.

In ESCs, the repression of bivalently marked genes that control lineage commitment and differentiation is partially regulated by EZH2[12,13]. In the context of cancer, Beguelin et al. previously showed that EZH2 suppresses differentiation of germinal centre B cells by establishing bivalent chromatin domains at promoters of differentiation and proliferation checkpoint genes[15]. To test whether differentiation genes are also held in a bivalent state in CC-ICs, we performed H3K4me3 and H3K27me3 ChIP-seq on POP92 CC-IC enriched cultures. We then assessed whether the expression of these bivalent genes was altered following UNC1999 treatment. This analysis identified 43 genes with H3K27me3-positive promoters (defined as 2.5 kb upstream and 0.5 kb downstream of the transcription start site) that were upregulated by UNC1999 treatment (Fig. 4a, Supplementary Figure 4c–e, Supplementary Table 3). A total of 20 promoters were bivalently marked (defined by the presence of directly overlapping H3K27me3 and H3K4me3 ChIP-seq signals), and a subset of 38 genes harboured a mixture of non-overlapping H3K27me3 and H3K4me3 regions within the promoter region (Fig. 4a, Supplementary Table 3). Conversely, none of the genes downregulated following UNC1999 treatment harboured H3K27me3 in their promoter. Gene set enrichment analyses combining the ChIP-seq and RNA-seq experiments show that promoters harbouring H3K27me3, mixed or bivalent marks, are more likely to be upregulated following UNC1999 treatment, with bivalent genes showing the strongest enrichment (Fig. 4c, Supplementary Figure 4f). In contrast, no significant enrichment was observed in promoters marked with H3K4me3 alone. Importantly, GO term analysis of the set of H3K27me3-marked and bivalent genes which are upregulated following UNC1999 treatment showed significant enrichment for cellular differentiation, which is in line with our observed sustained reduction in CC-IC self-renewal in vivo (Fig. 4d).

Prominent among these bivalently marked differentiation genes is Indian Hedgehog (*IHH*), showing a threefold increase in expression following UNC1999 treatment (Fig. 4d–f,

Supplementary Figure 4g). IHH is one of three Hedgehog pathway ligands in addition to Sonic (SHH) and Desert (DHH). IHH is the primary Hedgehog ligand expressed in the intestine, where it plays a key role in repressing the Wnt pathway and promoting differentiation[28,29]. Therefore, IHH was a strong candidate mediator of the decreased self-renewal capacity of CC-ICs following UNC1999 treatment. Using ChIP-qPCR, we confirm that the promoter of IHH is bivalently marked, and whereas H3K27me3 decreases following treatment with UNC1999, H3K4me3 remains unchanged (Supplementary Figure 4h–i). Comparing our list of bivalently marked genes with three published lists of bivalently marked genes from ESCs revealed that the promoter of *IHH* is also bivalently marked in ESCs[12,30,31]. This highlights a potential common epigenetic mechanism regulating lineage commitment for both CC-ICs and ESCs.

To assess whether any of the UNC1999-upregulated genes reflect activation of the IHH pathway, we assessed whether any DNA-recognition motifs for the HH pathway were enriched in *cis*-regulatory elements (CREs) of transcriptionally accessible chromatin in CC-IC cultures. ATAC-seq analysis of POP92 spheroid cultures revealed 35 CREs in accessible chromatin regions. Among these CREs, the DNA-recognition motif for GLI transcription factors, the downstream transcription factors of the HH pathway, was significantly enriched (~fourfold increase in abundance of the motif in active CREs versus the null expectation (Fig. 4g, Supplementary Data 2). Moreover, RUNX and SMAD DNA recognition motifs were also enriched in the CREs in open chromatin regions of CC-IC cultures (Fig. 4g). RUNX2 and SMADs transcription factors were previously shown to cooperate with IHH in other models[32]. SMAD1/5/8 are the downstream mediators of the bone morphogenic protein (BMP) pathway, which is known to cooperate in IHH-induced differentiation in intestinal stem cells[29]. Taken together, our data show that the IHH pathway, as well as known cooperators of IHH, are transcriptionally upregulated upon UNC1999 treatment in CC-IC enriched cultures.

**Exogenous IHH reduces self-renewal of CC-ICs.** IHH is a known regulator of normal intestinal and colon development, but its function in CC-ICs is unknown. To determine whether IHH plays a role in CC-IC self-renewal, we first confirmed our RNA-seq and ChIP-seq findings in three CC-IC enriched models. ChIP-PCR for H3K27me3 confirmed that the *IHH* promoter is repressed in all three models (Supplementary Figure 5a), and mRNA levels of IHH increased by two- to fourfold after 7 days of treatment with UNC1999 (Fig. 5a), while the other IHH ligands SHH and DHH, remain unchanged following treatment (Supplementary Figure 5b). To assess whether the increase in IHH results in activation of the HH pathway, POP92 cells were treated with either UNC1999 or using direct stimulation of the HH pathway using recombinant IHH. Both treatments led to an increase in *BMP4* and *GLI1* expression, two key Hedgehog target genes, suggesting that the UNC1999-induced increase in IHH also leads to increased activation of the HH pathway (Fig. 5b). Furthermore, markers of differentiation, such as FABP2[5] and CDX2, were upregulated following treatment with UNC1999 or recombinant IHH (Fig. 5c), whereas the stem cell markers Klf4, Oct4 and Nanog decreased in both IHH recombinant and UNC1999-treated CC-ICs, consistent with a decreased stem-like state (Fig. 5d). In line with these results, we observed a significant increase in CDX2 protein in UNC1999-treated xenografts compared with vehicle (Supplementary Figure 5c–d). Similar to our results with UNC1999, treatment with recombinant IHH also resulted in a decrease in Wnt reporter activity and Wnt target genes (Supplementary Figure 5e–g). This is consistent with a previous study looking at overexpression of IHH

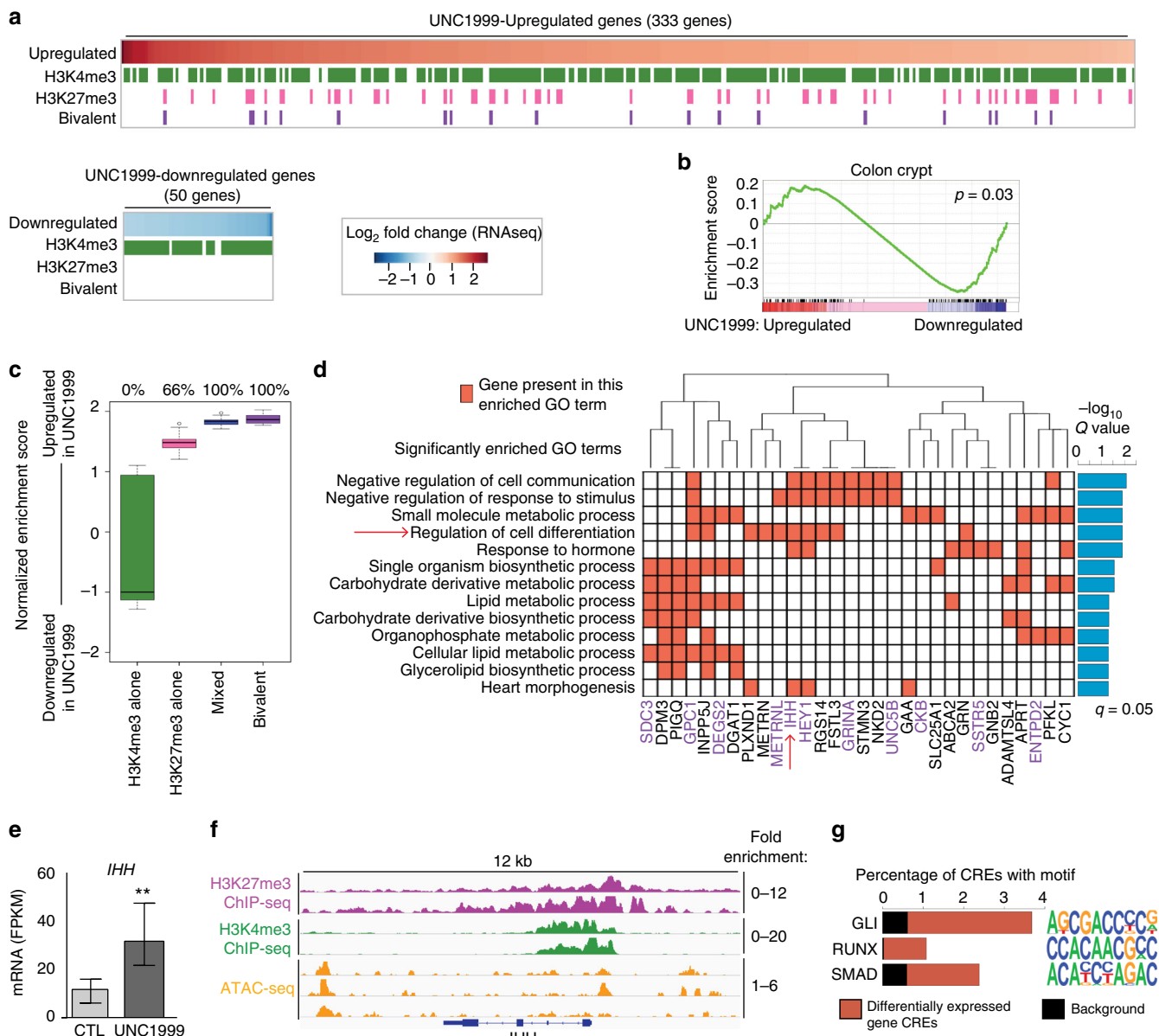

**Fig. 4** EZH2 Inhibition de-represses the colonic differentiation gene *IHH*. **a** RNA-seq heatmap of log$_2$ fold change for all genes significantly upregulated (red in top panel) or downregulated (blue in bottom panel) following UNC1999 treatment in POP92 cells. Lower bars underneath the heatmap indicate ChIP-seq data for POP92 cells in the absence of UNC1999. The presence of H3K4me3 peaks (green), H3K27me3 peaks (pink) or regions where both marks directly overlap (Bivalent; purple) in promoter regions (2.5 kb upstream of TSS, 0.5 kb downstream of TSS) is shown (prior to UNC199 treatment) for all genes whose expression is significantly changed after UNC1999 treatment (negative binomial, FDR-corrected q < 0.05). RNA-Seq was performed in (*n* = 3) biological replicates for each condition, ChIP-seq peaks called in *n* = 2 separate replicates were used for each mark. **b** GSEA performed on the differentially regulated transcripts from RNA-seq data with respect to the Colon Crypt signature (FWER, *p* < 0.03). **c** GSEA performed on 30 randomly selected sets of 200 genes with H3K4me3 (green), H3K27me3 (pink), mixed (blue) or bivalent (purple) marks in their promoters. Distribution of normalised enrichment scores are shown along with the percentage of runs which were significantly enriched among upregulated genes following UNC1999 treatment (*p* < 0.25, none were significant among downregulated genes). **d** Enriched biological process GO terms within all genes differentially upregulated following UNC1999 treatment with H3K27me3 marks in their promoter. Matrix shows which differentially expressed genes are present in each GO category, barplot shows −log$_2$ FDR-corrected q-value. Genes with bivalently marked promoters are highlighted in purple. **e** Boxplot of differential expression of IHH following UNC1999 treatment (mean of *n* = 3), error bars show max and min estimated FPKM values, Student's *t* test. **f** Representative ChIP-Seq tracks for H3K27me3 and H3K4me3 showing read counts over the IHH locus inclusive of the promoter region. **g** Transcription factor motif analyses performed on the predicted CREs (defined using ATAC-Seq) of UNC1999-differentially upregulated genes using homer, relative to a background of all ATAC-Seq peaks. **P < 0.01

in a CRC cell line[28]. These data combined with our in vitro and in vivo LDAs showing decreased CC-IC numbers confirm that CC-ICs are being driven to a more differentiated-like state following EZH2 inhibition.

To determine the effect of IHH on CC-IC self-renewal, we performed an LDA on three CC-IC enriched models pretreated with recombinant IHH (Fig. 5e). A significant ~1.8-fold decrease in CC-IC frequency was observed in the IHH pretreated group,

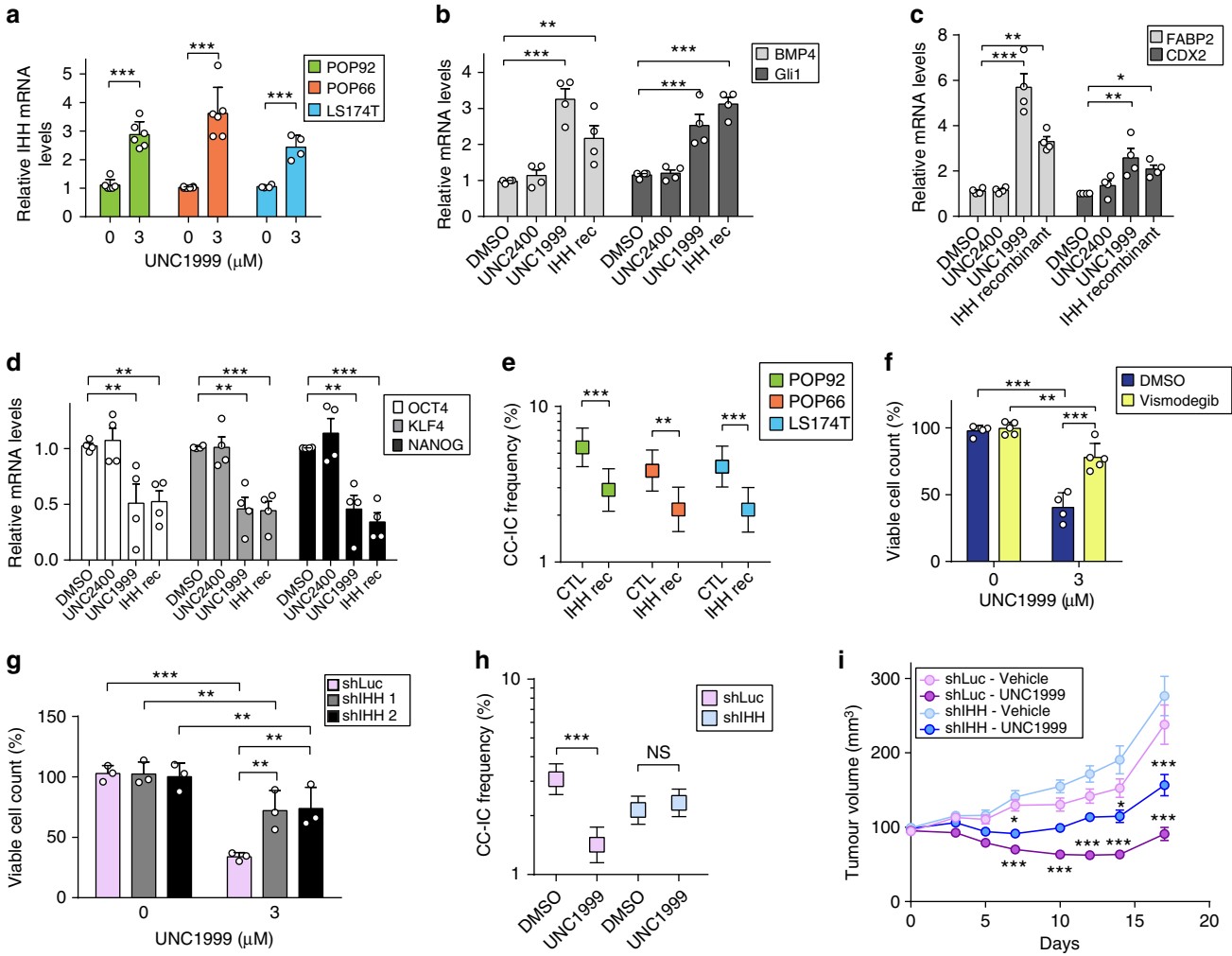

**Fig. 5** EZH2 regulates CC-IC self-renewal through control of IHH expression. **a** RT-qPCR for IHH of UNC1999-treated cells. Cells were treated with UNC1999 at 3 μM for 7 days and processed for RT-qPCR. Data shown are mean (POP66/POP92 n = 6, LS174T n = 4, ± SD, Student's t test). **b**–**d** mRNA Levels on POP92 CC-ICs of Hedgehog target genes Gli1 and BMP4 (**b**), differentiation markers FABP2 and CDX2 (**c**), and the stem cell markers OCT4, KLF4, NANOG (**d**) monitored by RT-qPCR following UNC1999 or UNC2400 treatment (3 μM), or recombinant IHH (5 μg/mL) for 7 days. Data shown are mean (n = 4, ± SEM, one-way ANOVA). **e** LDA on POP92, POP66 and LS174T cells pretreated with recombinant IHH (5 μg/mL) for 10 days. Data shown are n = 3 and represent mean ± 95% confidence interval, frequency and probability were computed using ELDA software. **f** POP92 spheroids treated with DMSO or UNC1999 (3 μM) in the presence of SMO inhibitor Vismodegib (10 μM) or DMSO control for 7 days. Viable cell count was measured and samples normalised to their respective control. Data shown are mean (n = 5, ± SD, two-way ANOVA). **g** IHH knocked down in POP92 using two different hairpins, and sensitivity towards UNC1999 was assessed using viable cell count via flow cytometry. Data are normalised to DMSO control performed in n = 3, ± SD, two-way ANOVA. **h** POP92 with IHH knockdown were treated with UNC1999 (3 μM) or DMSO control for 7 days prior to seeding in LDA. Data shown are n = 2 and represented as mean ± 95% confidence interval. Frequency and statistics were computed using ELDA software. **i** POP92 spheroids infected with either shRNA against Luciferase (shLuc) or shRNA targeting IHH (shIHH) were injected into mice and treated with vehicle or UNC1999 (300 mg/kg). Tumour volume was measured over time. Data are n = 20 tumours (shLuc ± UNC1999), n = 19 tumours (shIHH ± UNC1999), ± SEM, two-way ANOVA with Tukey multiple test correction. *P < 0.05, **P < 0.01, ***P < 0.001. Source data are provided as a Source Data file

providing functional evidence that IHH reduces the self-renewal capacity of CC-ICs.

**Inhibition of the HH pathway reduces the efficacy of UNC1999.** To further assess whether de-repression of IHH is one of the major consequences of EZH2 inhibition in our model, we co-treated POP92 spheroids with UNC1999 and Vismodegib, an FDA-approved Smoothened (SMO) antagonist, which inhibits the signalling of all three Hedgehog ligands SHH, IHH and DHH. Our data shows that inhibition of the HH pathway using Vismodegib reduces the efficacy of UNC1999 (Fig. 5f). Moreover, a 65% shRNA knockdown of IHH significantly rescued the effect of UNC1999 treatment on both the growth as well

as self-renewal capacity of CC-ICs in vitro, further supporting the predominant role of IHH in mediating the UNC1999 phenotype (Fig. 5g, h, Supplementary Figure 6a, b). Importantly, knockdown of IHH also reduces the efficacy of UNC1999 in vivo, limiting the decrease in tumour growth by 50% (Fig. 5i and Supplementary Figure 6b–e). Altogether, our results uncover *IHH* as a novel target gene held in a bivalent state by EZH2 in CC-ICs and provide evidence for a previously unobserved link between EZH2, regulation of *IHH* expression and CC-IC self-renewal. Furthermore, we show that CC-ICs remain responsive to normal cues of differentiation from IHH, which leads to decreased self-renewal capacity and increased differentiation.

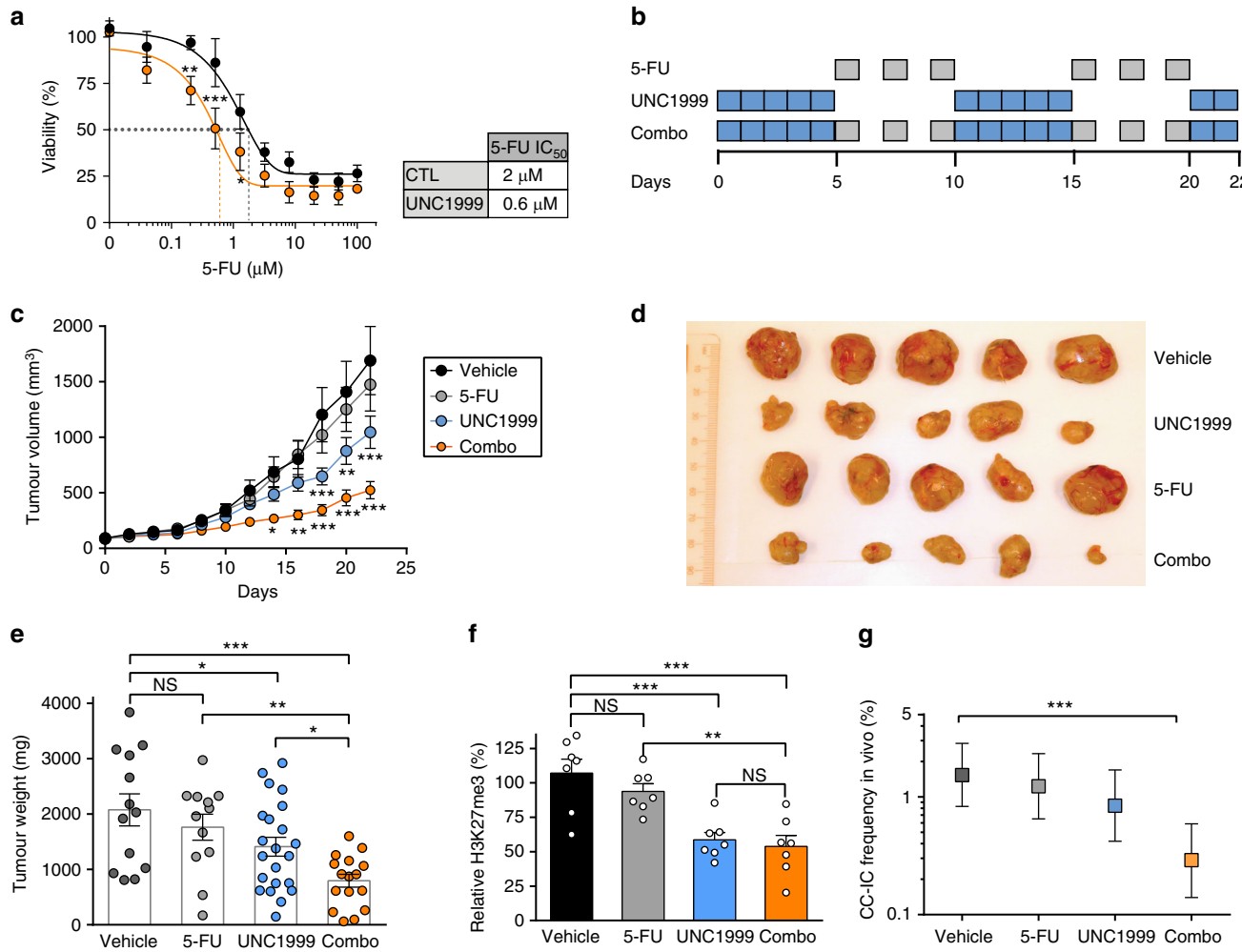

**Fig. 6** UNC1999 treatment in vivo increases chemosensitivity to 5-Fluorouracil. **a** POP92 in vitro combination of UNC1999 (3 μM) and increasing doses of 5-FU. Cells were pretreated with UNC1999 (3 μM) for 3 days, after which 5-FU was added and cells incubated for 4 additional days. Viable cell count was measured and samples normalised to their respective control. Data shown are mean ($n = 4$, ±SEM, two-way ANOVA). Table shows shift in 5-FU IC$_{50}$ in UNC1999-pretreated cells. **b** Sequential treatment of UNC1999 (300 mg/kg, oral gavage) and 5-FU (15 mg/kg, IP). Mice were dosed with either vehicle, UNC1999 alone, 5-FU alone, or 5-FU and UNC1999 in combination (Combo): 10-day cycles starting with UNC1999 for 5 days, followed by 5-FU every other day for 5 days. **c** Tumour size was measured over time. Data represent mean ± SEM of $n = 13$ tumours (vehicle), $n = 12$ (5-FU), $n = 22$ (UNC1999) or $n = 15$ (combo), two-way ANOVA compared to vehicle alone. **d** Xenografts collected from the mice treated in (**c**). **e** Tumour weight measurements of the xenografts collected in (**c**) and shown in (**d**). Data represent mean ± SEM of $n = 13$ tumours (vehicle), $n = 12$ (5-FU), $n = 22$ (UNC1999) or $n = 15$ (combo), one-way ANOVA. **f** Quantification of the reduction in H3K27me3 by western blot. Data shown are $n = 7$ tumours ± SEM, one-way ANOVA. **g** In vivo LDA performed with xenografts collected in (**c**). Data shown are represented as mean ± 95% confidence interval. Frequency and statistics were computed using ELDA software. \*$P < 0.05$, \*\*$P < 0.01$, \*\*\*$P < 0.001$. Source data are provided as a Source Data file

**UNC1999 treatment in vivo increases chemosensitivity.** Chemoresistance of CC-ICs has emerged as an important cellular property that enables tumours to recur following cytoreductive therapy[33]. We hypothesised that the decrease in CC-IC frequency and more differentiated state of the CC-IC enriched cultures following UNC1999 would result in increased sensitivity to 5-fluorouracil (5-FU), a standard of care chemotherapeutic agent in CRC. In agreement, we observe a threefold increased sensitivity towards 5-FU in the UNC1999-pretreated CC-IC enriched models in vitro (Fig. 6a). We carried out an in vivo study using a sequential dosing regimen starting with 5 days of UNC1999 to allow epigenetic reprogramming to occur, followed by 5-FU every other day for 5 days, using sub-optimal dosing regimen for both treatments to prevent toxicity in mice (Fig. 6b and Supplementary Figure 7a). No significant change in tumour growth was observed using 5-FU alone at 15 mg/kg, while UNC1999 decreased tumour volume by 40%. However, the combination of UNC1999 and

5-FU resulted in a 70% decrease in tumour volume and weight (Fig. 6c–e). A 40% reduction of H3K27me3 was observed in both UNC1999 alone and co-treated tumours (Fig. 6f and Supplementary Figure 7b). Analysis of the tumours show large necrotic areas in the combination group, as well as increased cleaved Caspase 3 staining (Supplementary Figure 7c).

To determine the effect of combination therapy on the CC-IC fraction, we carried out an in vivo LDA using xenograft cells from mice co-treated with UNC1999 and 5-FU. This revealed a fourfold decrease in the tumour-initiating capacity in cells isolated from co-treated tumours compared to control or 5-FU alone, and a 1.8-fold decrease for the sub-optimal dosing of UNC1999 (Fig. 6g). Therefore, combining 5-FU with UNC1999 further decreases tumour growth and increases the elimination of the CC-IC fraction. Moreover, xenograft mRNA analysis showed an increase in IHH as well as the differentiation markers FABP2 and CDX2 in the UNC1999 group but not in the combination,

suggesting that the sequential dosing of UNC1999, followed by 5-FU, pushes CC-ICs towards a differentiated-like state, which in turn are targeted by 5-FU (Supplementary Figure 7d–f).

## Discussion

EZH2 has an established role in regulating pluripotency and maintaining self-renewal in ESCs through repression of lineage commitment and differentiation genes that are maintained in a bivalent state[12–14]. Loss of the repressive histone mark H3K27me3 at bivalent genes results in the rapid transcriptional initiation of lineage-specific differentiation programmes, indicating that ESCs are poised to undergo differentiation[12–14]. Although much is published on the role of EZH2 in maintaining C-ICs in a broad range of solid tumours, the mechanisms identified are diverse, with most publications attributing the effect on self-renewal to methylation of non-histone targets[34,35], recruitment of DNA methyltransferases[36] and/or direct binding to other proteins[37,38]. Whether EZH2's canonical role in ESCs contributes to maintaining the C-IC state has never been established. Our results uncover an epigenetic block to differentiation in CC-ICs that can be overcome by targeting the bivalent state of key developmental genes. Furthermore, we functionally demonstrate that CC-ICs retain the capacity to respond to a normal cue of cell fate specification mediated by IHH, a key driver of normal colonocyte differentiation. These findings have broad implications for the C-IC field as we demonstrate a way to target C-ICs through de-repression of bivalently marked differentiation genes. Our results highlight the importance of identifying other, tissue-specific bivalently marked differentiation genes in C-ICs from other disease models, and how re-expression of these genes through epigenetic modulation could trigger C-ICs to exit the stem-like state.

Literature on the role of bivalency in C-ICs remains very limited. Initial work studying plasticity in human basal breast C-ICs demonstrated that non-C-ICs are plastic populations that can switch to a C-IC state through maintaining the ZEB1 promoter in a bivalent configuration. In response to TGFβ, the ZEB1 promoter converted from a bivalent to active chromatin configuration, and as a result non-C-ICs converted to C-ICs[39]. Aiden et al. showed that poorly differentiated Wilms tumours share an epigenetic landscape with ESCs, observing similar bivalent chromatin modifications at a subset of promoters in both cell types[40]. In ovarian cancer, bivalent genes that were identified in ESCs and one ovarian tumour sample were found to be expressed at lower levels in ovarian C-ICs compared with non-C-ICs, where C-ICs were defined by the side-populations[31]. Although the authors showed that bivalent genes in ESCs are more repressed in C-ICs, the potential functional consequence of the increased repression was not explored. To date, our work is the first study to functionally demonstrate that the bivalent state of key developmental gene promoters maintains C-ICs locked in an epigenetically maintained stem-like state.

Genome-wide profiling of ESCs and cancer cell lines or bulk tumour samples shows that bivalently marked promoters of differentiation genes in ESCs gain aberrant DNA hypermethylation in CRC cell lines[41,42]. This suggests that these differentiation genes are permanently silenced in CRC, which would render loss of H3K27me3 irrelevant. However, we demonstrate that C-ICs retain the capacity to exit the stem-like state upon inactivation of EZH2 resulting in the loss of H3K27me3. One possible explanation for the conflicting results is that cancer cell lines grown in adherent conditions in the presence of serum show significantly fewer bivalent regions than normal tissue or ESCs[43], suggesting that our findings on the role of bivalency in CC-IC self-renewal may have been made possible through the use of patient-derived,

CC-IC enriched 3D spheroid/organoid models. It is also plausible that the role of DNA methylation in permanently silencing bivalently marked promoters may be tumour-specific or associated with poorly differentiated cancers. Therefore strategies that combine EZH2 inhibitors together with DNA methylation inhibitors may be beneficial to optimally target silenced differentiation programmes[44].

IHH is one of the canonical functional markers responsible for colonocyte differentiation. In the adult colon, IHH is expressed by terminally differentiated colonocytes, and has been shown to restrict Wnt signalling to the bottom of the colon crypts as well as restricting the size of the crypts[28]. Furthermore, specific knockout of Ihh in murine models promotes the expansion of the stem cell compartment, indicating that Ihh plays a central role in driving intestinal stem cell differentiation[29]. Several reports studying IHH in the context of normal small intestinal and/or colon tissue, provide evidence that HH signalling is paracrine, where the stromal cells respond to IHH produced by the enterocytes[29,45–47]. However, recent reports provide evidence for cell-autonomous HH signalling in CRC models[48–51]. Here, using patient-derived CC-IC-enriched models, we show that increased IHH expression through treatment with UNC1999, or direct stimulation with recombinant IHH, reduces self-renewal capacity of CC-IC enriched cultures even in the absence of stroma. The conflicting observations around IHH signalling in normal colon and CRC might arise from the various models used in the studies, such as normal small intestine or colon, different tumour stages, and cancer cell line models. It is also likely that a unique subset of cells within a tumour may respond differently to IHH, such as the CC-IC subset. When we antagonise canonical HH signalling through treatment with Vismodegib, we observe a dramatic reduction of the efficacy of UNC1999, which demonstrates that HH signalling is a key consequence of EZH2 inhibition in patient-derived CC-ICs. In line with our observations, it has been shown that treatment with Vismodegib or IHH knockout mouse models lead to increased tumour occurrence and larger tumours[47]. A recent clinical trial was carried out in CRC using Vismodegib, and showed no added benefit over standard-of-care alone[52]. Our data supports a distinct role for IHH in driving CC-ICs to exit the stem-like state. Thus, our results may explain, in part, the failure of Vismodegib in clinical trials of CRC; Vismodigib prevents cellular signalling by all three Hedgehog ligands including IHH, which based on our findings, would favour the CC-IC state.

In our models, targeting bivalency through EZH2 inhibition significantly reduced CC-IC self-renewal; however, we note that it did not fully eradicate the CC-IC population. One potential reason for this could be incomplete removal of the H3K27me3 mark in tumour cells using UNC1999, as we observed that the decrease in tumour size was approximately proportional to the decrease of the mark. Therefore, pharmacological agents with greater exposure at the tumour may be more efficacious. We also demonstrate that in vivo treatment with UNC1999 followed by 5-FU, resulted in a statistically significant reduction in tumour growth and CC-IC frequency, as compared with either compound given alone. EZH2 inhibitors are emerging from phase I studies, none of which included CRC, and there is great interest in identifying the best patient populations and clinical strategies for their future clinical development. Our work shows that EZH2 inhibitors could be incorporated as a part of novel adjuvant therapy combinations in the context of CRC. We hypothesise that by treating with UNC1999, IHH was de-repressed, resulting in differentiation of CC-ICs and as a result increased sensitivity to 5-FU. Currently there is a paucity of literature on whether bivalency is playing a role in determining response to chemotherapy. However, a study published in ovarian cancer showed that bivalently marked ESC genes are expressed at much lower levels

in chemo-resistant ovarian cancer cells as compared with matched chemo-sensitive cells, albeit the functional significance of this finding remains to be determined[31]. A better understanding of the effect of disrupting bivalent promoters in C-ICs in response to standard of care chemotherapy could lead to novel therapeutic strategies.

Taken together, our results identify that similar to ESCs, CC-ICs maintain key lineage specification genes in a bivalent state. Moreover, we functionally demonstrate that CC-ICs remain responsive to a normal differentiation cue, IHH, upon loss of H3K27me3 or direct stimulation with recombinant IHH. We have identified a novel means to therapeutically target the CC-IC fraction through disrupting the bivalent state of key differentiation genes. Our findings will lead to novel therapeutic strategies aimed at identifying and targeting bivalently marked differentiation genes in cancer, and specifically C-ICs.

## Methods

**Colorectal cancer patient-derived xenografts**. Human CRC tissue was obtained with informed patient consent, as approved by the Research Ethics Board at the University Health Network (UHN) in Toronto, Canada. To establish and maintain PDX models, cells from freshly dissociated CRC tissue or freshly thawed cells previously frozen xenograft samples[53] were mixed (1:1) with high concentration Matrigel (Corning) and injected subcutaneously into the flanks of NOD-SCID mice (male, 6–8 weeks of age). All animal experiments were reviewed and approved by the Animal Care Committee at the University Health Network in Toronto.

**Spheroid CC-IC enriching cell culture**. Human CRC tissue was obtained with informed patient consent, as approved by the Research Ethics Board at the University Health Network in Toronto. Patient-derived spheroid lines were established from xenografts[17] or directly from patient samples at the time of surgical resection[17]. Samples were dissociated, depleted for mouse cells (applicable to xenografts) and cultured in suspension using serum-free growth factor enriched media as spheroids, in suspension culture flasks at 37 ºC in a 5% CO2-humidified incubator as previously described[7]. The culture medium contained DMEM/F-12 with GlutaMAX (1:1 ratio) (Thermo Fisher), supplemented with penicillin–streptomycin (1%) (Thermo Fisher), nonessential amino acids (1 ×), sodium pyruvate (1 mM), N2 supplement (STEMCELL), NeuroCult SM1 Neuronal Supplement (STEM-CELL), heparin (4 µg/ml, Sigma), lipids (0.2%, Sigma), EGF (20 ng/ml, Reprokine) and basic FGF (10 ng/ml, Reprokine). All cell lines were authenticated using short tandem repeat profiling, and proven to be negative for mycoplasma. Spheroids were kept in culture for a maximum of 8–10 passages.

**Organoid culture**. Fresh CRC tumour tissue was obtained with informed patient consent, as approved by the Research Ethics Board at the University Health Network in Toronto. Fresh tumour tissue was cut into small pieces and digested with Liberase (TH grade; Roche) for 90 min at 37 °C, and dissociated cells were collected and embedded in growth factor-reduced Matrigel (Corning). The Matrigel and embedded cells were overlaid with growth medium containing advanced DMEM/F-12 (Gibco) supplemented with 2 mM GlutaMAX (Gibco), 10 mM HEPES (Gibco), 100 U/ml penicillin–streptomycin (Gibco), 1 × B-27 Supplement (Gibco), 1.25 mM N-acetyl-L-cysteine (Sigma), 10 nM Gastrin I (Sigma), 50 ng/ml mouse EGF (Gibco), and 500 nM A83-01 (Tocris). All organoid models were authenticated using short tandem repeat profiling, and proven to be negative for mycoplasma. Organoids were kept in culture for a maximum of 8–10 passages.

**Epigenetic probes screen and viable cell count assays**. The Epigenetic probe library was provided by the Structural Genomics Consortium. For the POP92 spheroid epigenetic probes screen, cells were seeded at a density of 10,000 cells per well in a non-treated U-bottom 96-well plate (Sarstedt) and subsequently treated with the appropriate compounds from the Epigenetic probe library (Structural Genomics Consortium), with recombinant IHH (5 µg/mL, Biolegend), or Vismodegib (10 µM, ApexBio). After 7 days, the cells were trypsinized and stained with Sytox Blue viability dye (Thermo Fisher) for viable cell count using flow cytometry. Plates were acquired on a BD LSR II with a High-Throughput Sampler unit (BD Biosciences) and on MacsQuant VYB with plate adapter (Miltenyi Biotec). Data were analysed using FlowJo v10.

For the organoid epiprobe screen, organoids were dissociated to single cells using TrypLE Express (Thermo Fisher). Tissue culture-treated 384-well plates with white wall and transparent bottom were coated with 5 µL of Matrigel (Corning), and allowed to polymerise at 37 °C for 20 min, before seeding 1000 cells per well in organoid growth media containing 2% Matrigel. Twenty-four hours after seeding, compounds diluted in organoid growth media containing 2% Matrigel were added to the cells. The plates were incubated at 37 °C for 7 days and processed for CellTiterGlo-3D (Promega) following the manufacturer's instructions. Plates were

read using a CLARIOstar plate reader (BMG Labtech). Data were normalised to DMSO control.

**Western blotting**. Cells were lysed in ice-cold lysis buffer (20 mM Tris–HCl pH 8, 150 mM NaCl, 10 mM MgCl2, 1 mM EDTA, 0.5% Triton X-100 + protease inhibitors) for 15 min at 4 °C. Samples were then sonicated in a Q800 Waterbath sonicator (QSonica) for 15 cycles of 8 s ON, 15 s OFF. Lysates were cleared after 15 min centrifugation at 10,000 g, and protein concentration was determined using Pierce BCA Protein assay kit (Thermo Fisher). Protein extracts were resolved by SDS-PAGE, blotted to PVDF membranes and probed using the following antibodies: anti-EZH2 (Cell Signaling Technologies, Cat nb#5246, 1:1000), anti-H3K27me3 (Diagenode, Cat nb# C15200181-50, 1:5000), anti-H3K27me2 (Abcam, Cat nb# ab24684, 1:1000), anti-H3K27me1 (Active Motif, Cat nb# 61015, 1:1000), anti-H3 (Abcam, Cat nb# ab1791, 1:2000), GAPDH (Millipore, Cat nb# MAB274, 1:10,000), actin (Abcam, Cat nb# ab3280, 1:5000). Signal was detected using IRDye 680RD and 800CW secondary antibodies following the manufacturer's instructions (LI-COR). Membranes were acquired on an Odyssey Imager (LI-COR) and quantified using Image Studio (LI-COR). Uncropped images of all western blots are provided in the Source Data file.

**Cell cycle and apoptosis/necrosis**. For PI cell cycle analysis, cells were pretreated for 7 days with UNC1999 at 3 µM, trypsinized, washed and fixed in ice-cold ethanol, washed twice in PBS and resuspended in propidium iodide (50 µg/mL, Sigma) supplemented with RNase (100 µg/mL). For EdU/Hoechst cell cycle analysis, trysinized POP92 cells were incubated with 10 µM EdU (Invitrogen) for 2 h at 37 °C. Cells were then fixed with 4% PFA and Click-IT Plus Alexa Fluor 488 flow cytometry kit (Invitrogen) was used as per the manufacturer's instructions. Hoechst 33342 was added at a concentration of 20 µg/mL (Invitrogen) for 30 min. Cells were analysed using MACSQuant VYB flow cytometer (Miltenyi) and analysed with FlowJo (version 10) software. Apoptosis/necrosis using Annexin V-FITC and propidium iodide was performed following the manufacturer's instructions (Annexin V-FITC apoptosis detection kit, Sigma). Cells were acquired on a BD LSR II (BD Biosciences).

**TCF/LEF GFP Wnt reporter assays**. POP92 and POP66 were infected with TCF/LEF promoter-driven GFP reporter lentivirus following the manufacturer's instructions (Qiagen), or with FOP-mutated TCF/LEF negative control[54] (Addgene). Reporter cells were pretreated for 7 or 10 days with UNC1999 or GSK343 at 3 µM, or recombinant IHH (5 µg/mL) (Biolegend) for 10 days. Cells were trypsinized, washed and resuspended in PBS with Sytox blue viability dye, and acquired using the MacsQuant VYB (Miltenyi Biotec). Data were normalised to the top 10% brightest population in the DMSO control.

**Tissue microarray and immunohistochemistry**. The patient tissue microarray slides were generated by the University Health Network Biobank, obtained with informed patient consent, as approved by the Research Ethics Board at the University Health Network in Toronto. TMA and xenograft slides were stained using EZH2 antibody (Cell Signaling Technologies, Cat nb#5246, 1/100 dilution) and H3K27me3 (Cell Signaling Technologies, Cat nb#9733, 1/200 dilution) following the manufacturer's instructions. The slides were scanned using a 3DHistech Pannoramic 250 Flash II slide scanner using a ×40 objective. The IHC score of the TMAs was assessed manually and confirmed by a pathologist. The percentage of positive cells (P) was assessed and scored as follows: 0 (0–10% positive cells); 1 (10–25% positive cells); 2 (26–50% positive cells); 3 (51–75% positive cells); and 4 (≥ 76% positive cells). The overall staining intensity (I) was scored as: 0 (negative), 1 (weak), 2 (moderate) and 3 (strong). The final IHC score was calculated by multiplying the quantity of positive cells P (0-4) by the staining intensity I (0–3), ranging from 0 to 12.

Cleaved Caspase 3 (Cell Signaling Technologies, Cat nb#9661, 1/600 dilution), CDX2 (Dako M3636, 1/50) and Ki67 (Novus, Cat nb# NB110-90592, 1/2000 dilution) were used following the manufacturer's instructions. Quantification of Ki67 was performed by manual counting of DAB-positive cancer cell nuclei over total cancer cell nuclei per 40x field, 5–6 fields per tumour.

**Limiting dilution assays in vitro**. Cells were pretreated with 3 µM of UNC1999 or recombinant IHH (5 µg/mL) (Biolegend) for 10 days, before being trypsinized and seeded in serial dilutions of 1000, 100, 10 and 1 cell per well of a 96-well plate. Cells were incubated for 4–6 weeks at 37 °C in the absence of additional treatment, and every well was assessed for presence or absence of sphere. The data were analysed using the online Extreme Limiting Dilution Analysis (ELDA) Bioinformatics tool[55].

**RT-qPCR**. RNA was extracted using RNeasy kit (Qiagen) following the manufacturer's instructions. cDNA was generated using iScript Advanced (Biorad), and qPCR was set up using PowerUP SYBR green qPCR mastermix (Thermo Fisher) and run on a CFX384 Touch Real-Time PCR detection system (Biorad). Housekeeping genes TBP and/or 18S were used to normalise the data. Primer sequences are listed in Supplementary Table 4.

**shRNA infection.** shRNA hairpins for Control, EZH2 and IHH shRNA knock-down were obtained from Sigma (Mission shRNA TRCN0000033320 (IHH sh1), TRCN0000033321 (IHH sh2), TRCN0000040077 (EZH2 sh1), TRCN0000018365 (EZH2 sh2), TRCN0000072246 (Luciferase CTL shRNA)). Lentivirus were generated in HEK293, collected and used for infection of spheroid cultures pre-treated with 8 µg/mL Polybrene (Sigma). Twenty-four hours after infection, cells were washed and allowed to recover before selection using Puromycin dihydrochloride (Sigma). After selection, cells were processed for the indicated experiments.

**In vivo UNC1999 dosing and limiting dilution assays.** All animal experiments were reviewed and approved by the Animal Care Committee at the University Health Network in Toronto. For UNC1999 in vivo dosing experiments, UNC1999 was synthesised in the Jin Laboratory. CB.17 female scid/scid mice were injected with $3 \times 10^5$ CC-IC enriched cells subcutaneously at two tumours per mouse. UNC1999 was prepared at 300 mg/kg in 0.5% carboxymethylcellulose and 0.1% Tween, and administered to the mice when the tumours reached around 150 mm³. The treatments were performed once daily by oral gavage, for 9 consecutive days and a 2-day break, for a total of 21 to 25 days (until vehicle reached maximum size), or for 5 consecutive days and 5 days break for the combination experiment in Fig. 6. Body weights were measured every day over the course of treatment, and tumour growth was monitored by caliper measurements every 2–3 days until endpoint was reached.

For in vivo LDA, tumours were collected from the UNC1999 dosing experiments, digested, counted for viable cells using Trypan blue exclusion, and injected into NOD/SCID γ (NSG) mice at a limiting dose (10000, 1000, 100 and 10 cells per injection). Tumour formation was monitored over the course of 4 months, and stem cell frequencies determined using the online Extreme Limiting Dilution Analysis bioinformatics tool[55].

**ChIP-Seq, ChIP-qPCR and RNA-seq.** ChIP was carried out with POP92 spheroids. Briefly, Dynabeads A and G were incubated with anti-H3K27me3 antibody (Abcam ab6002), anti-H3K4me3 antibody (Abcam ab8580), rabbit IgG Isotype control or mouse IgG Isotype control for 6 h. In the meantime, cells were fixed with 1% formaldehyde for 10 min, washed with PBS/1% BSA and then with PBS. Fixed cells were lysed in lysis buffer (1% SDS, 10 mM EDTA, 50 mM Tris–HCl pH 8.1), incubated on ice for 10 min and sonicated using a 4 °C waterbath sonicator Bioruptor (Diagenode). Cells were sonicated for 60 cycles of 30 s ON and 30 s OFF, to generate fragments between 200–300 bp. Lysates were clarified by centrifugation at 16,000 g, 4 °C for 15 min, and the supernatants were transferred to new tubes. Ninety percent of the cross-linked chromatin was incubated overnight along with antibody–dynabeads complexes, whereas 10% was stored at 4 °C as an input. Thereafter, samples were washed and decross-linked overnight with a decross-linking buffer (1% SDS and 0.1 M NaHCO₃) at 65 °C. Samples were purified using MinElute PCR purification kit (Qiagen) and processed for qPCR or sequencing, respectively. The qPCR experiments were set up using PowerUP SYBR green qPCR mastermix (Thermo Fisher) and run on a CFX384 Touch Real-Time PCR detection system (Biorad). Primer sequences are listed in Supplementary Table 4. For sequencing, fragments from 240–360 bp were selected using the PippinHT system (Sage Science). Libraries were prepped using the ThruPLEX DNA-Seq Kit, following the manufacturer's instructions (Rubicon Genomics), and sequenced using Illumina HiSeq 2500 sequencer, V4 chemistry.

RNA-seq was performed on POP92 cells pretreated with UNC1999 at 3 µM for 7 days. Cells were collected and RNA was extracted using Qiagen RNeasy kit. RNA concentration and quality was assessed using a Bioanalyzer (Agilent). Sample library preparation was performed using Illumina TruSeq Stranded mRNA sample preparation kit. Sequencing was performed on Illumina NextSeq500 using 75-cycle paired-end protocol and multiplexing.

**Assay for Transposase-Accessible Chromatin (ATAC-Seq).** To determine the chromatin accessibility of POP92 cells, an Assay for Transposase-Accessible Chromatin with high-throughput sequencing (ATAC-Seq) was performed. In total, 30,000 live cells were washed with PBS and lysed for 5 min on ice using a lysis buffer containing 10 mM Tris–HCl, pH 7.4, 10 mM NaCl, 3 mM MgCl₂, 0.1% IGEPAL CA-630. After isolating crude nuclei, samples were treated for 30 min at 37 °C with Tn5 transposase, and then the DNA was purified by MinElute PCR Purification Kit. Transposed DNA fragments were amplified using specific adapters followed by purification with MinElute PCR Purification Kit. Fragments from 240–360 pb were selected in the PippinHT system. The quality of the library and its DNA concentration were assessed by Bioanalyzer instruments and sequenced using Illumina HiSeq 2500 sequencer, V4 chemistry.

**NGS data processing and analysis.** RNA-Seq reads were aligned to the hg19 human reference genome using Bowtie[56] (2.0.5) and Tophat[57] (2.0.8) using default settings. Cufflinks[58] (2.1.1) was used to compute and normalise read counts, and call differentially expressed genes. ChIP-Seq and ATAC-Seq reads were aligned to the hg19 human reference genome using bwa using default settings, all reads with a quality score of less than 30 were removed, along with all reads mapped to regions marked as belonging to the hg19 blacklist[59], chrM, chrY and unlocalized sequences.

Picard tools (http://broadinstitute.github.io/picard/) (1.84) was used to mark all duplicate reads. MACS[60] (2.0.1) was used to call enriched peaks and calculate the observed fold enrichment over background. Peaks identified in both of two ChIP-Seq replicates for each condition were retained. Promoter areas as defined by −2.5 kb − + 0.5 kb from the TSS, were first identified. Bivalent promoters were then defined as those having H3K4me3 and H3K27me3 ChIP-seq peaks that directly overlapped within the promoter region. Promoters that contained both H3K4me3 and H3K27me3 peaks that did not overlap were classified as "mixed".

The GSEA tool[61] was used to assess whether UNC1999 treatment decreased enrichment for the Colon Crypt signature. Overall, 10,000 permuations of gene sets were used to compare the UNC1999 and DMSO treated samples. P-values were corrected for family-wise error rate. GSEA was also used to compute whether genes with H3K27me3, H3K4me3 or both marks present at their promoters were enriched in one RNA-Seq condition. Due to the large number of genes harbouring each mark, 30 random subsets of 200 genes were obtained and GSEA performed on each subset. The distribution of normalised enrichment scores was plotted as a boxplot. The least significant GSEA result was presented for each, as well as the distribution of enrichment scores.

For all differentially expressed genes, potential cis-regulatory elements (CREs) associated with those genes were identified via correlation between their promoters and DNASE hypersensitive sites via Thurman et al.[62]. The catalogue of all merged CREs called as accessible in any ATAC-Seq sample were considered putative CRC-accessible CREs. Homer[63] (4.9) was used to identify enriched transcription factor recognition motifs at these putative CRC-accessible CREs against a background of all ATAC-Seq peaks called in at least one replicate.

**Statistical analyses.** All measurements were taken from distinct samples/biological replicates analyses. With the exception of the Limiting Dilution assays and epigenomics analyses, the statistical analyses were performed using Graphpad Prism 6. For LDAs (Figs. 2h–j, 3i, j, 5h, 6g the statistical significance were obtained through ELDA bioinformatics tool[55]. Unless otherwise indicated in the figure legends, the statistics analyses were performed as follows: one-way ANOVA with Dunnet multiple test correction (Fig. 1a), one-way ANOVA with Bonferroni multiple test correction (5c, 6f, Supplementary Figure 2g, Supplementary Figure 4g, Supplementary Figure 5g, Supplementary Figure 6a, Supplementary Figure 6c, d, Supplementary Figure 7d–f). Two-way ANOVA with Bonferroni multiple test correction (Figs. 1c,1e, 1g, 2f, 2b, c, 2e, 2g, 5a,b, 5d, 5f, 6c, Supplementary Figure 1b, Supplementary Figure 1e, Supplementary Figure 2a, Supplementary Figure 2c, Supplementary Figure 2h, Supplementary Figure 5a, b). All Student's t test were performed as two-tailed (Figs. 1d, 2b, Supplementary Figure 3c, d, Supplementary Figure 5d).

Statistical analysis of qPCR data was performed in GraphPad Prism. Values were obtained from at least three biological replicates as indicated in each respective figure legend, performed in four technical replicates and were normalised to TBP and/or 18S housekeeping genes.

**Ethics.** Approval for this study was given by the Research Ethics Board at the University Health Network (UHN), Toronto, Canada (REB 10-0705-TE to CAOB). Patient tissue was obtained by UHN with informed patient consent as per UHN's Research Ethics Board guidelines. This study complies with all the relevant regulations for animal testing and research, as reviewed and approved by the Animal Care Committee at the University Health Network in Toronto, Canada (AUP 2781 to CAOB).

## Data availability

All the data for RNA-seq, ATAC-seq and ChIP-seq are available through GEO (peak tracks, GSE113176 and in the European Genome-Phenome Archive (raw sequencing data, EGAS00001003003. Source data underlying Figs. 1–3, 5–6, and Supplementary Figures 1–7 are provided as Source Data file. Gating strategy for Flow Cytometry experiments, as well as uncropped images of Western Blots are provided in the Source Data file. A reporting summary for this Article is available as a Supplementary Information file. All the patient samples (tumours and organoids) are available through the Princess Margaret living biobank upon request.

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

## Acknowledgements

We would like to thank Drs. John E. Dick, Laurie Ailles, as well as all the members of the Lupien & O'Brien labs for critical review of the manuscript and constructive discussions. We also thank Fannong Meng, Mayleen Sukhram, and Dianne Chadwick (UHN Biospecimen Sciences Program) for providing colorectal cancer samples as well as Tissue Microarray guidance. E.L.-F. is the recipient of a postdoctoral fellowship by the National Research Fund and the Marie Curie Actions of the European Commission (FP7-COFUND), and subsequently of a Canadian Institutes of Health Research (CIHR) Banting Postdoctoral Fellowship. This work was supported by CIHR (FRN-125792 to C.H.A., C.A.O.B. and M.L.), the U.S. National Institutes of Health (R01CA218600, R01CA230854, R01GM122749 and R01HD088626 to J.J.). The Structural Genomics Consortium is a registered charity (number 1097737) that receives funds from AbbVie, Bayer Pharma AG, Boehringer Ingelheim, Canada Foundation for Innovation, Eshelman Institute for Innovation, Genome Canada through Ontario Genomics Institute [OGI-055], Innovative Medicines Initiative (EU/EFPIA) [ULTRA-DD grant no. 115766], Janssen, Merck KGaA, Darmstadt, Germany, MSD., Novartis Pharma AG, Ontario Ministry of Research, Innovation and Science (MRIS), Pfizer, São Paulo Research Foundation-FAPESP, Takeda, and the Wellcome Trust.

## Author contributions

C.H.A., C.A.O.B., M.L., D.D.D.C., E.L.-F. and A.M. designed the project. E.L.-F., C.H.A. and C.A.O.B. designed experiments, analysed and interpreted the data. A.M., M.L., D.B.-L. and D.D.D.C. assisted in data interpretation. E.L.-F. performed the majority of experiments, Y.W. carried out the in vivo studies, T.M. performed the ChIP-seq experiments, and G.M.L. performed EdU/Hoechst cell cycle experiments. S.D., C.Z. and C.L. provided technical assistance. A.M. performed the bioinformatics analyses. A.P. performed IHC necrosis quantification. A.Ma and J.J. provided UNC1999 for the study. J.H. and B.G.W. provided patient-derived organoid samples. C.H.A., C.A.O.B. and E.L-F. wrote the paper. A.M., M.L., D.D.D.C. and J.H. provided critical evaluation of the paper. All authors read and approved the final version of the paper.
