## [Peer Review File · Nature Communications]

Reviewers' comments:

Reviewer #1 (Remarks to the Author):

According to the cancer stem cell model, a subset of tumour cells, referred to as Cancer Stem Cells (C-ICs), have a capacity for self-renewal and tumour initiation. Polycomb-group (PcG) proteins contribute to the maintenance of the stem cell state by maintaining the transcriptional repression of differentiation-promoting genes. The promoters of these genes typically are bivalently marked with the antagonistic H3K4me3 and H3K27me3 epigenetic tags. EZH2, the catalytic component of the PcG PRC2 complex is responsible for depositing H3K27me3, which leads to recruitment of PRC1 and transcriptional repression. EZH2 inhibitors have been developed and currently are being tested in multiple cancer clinical trials. In this study, the authors investigate the role of PcG-mediated repression in maintaining the proliferative capacity of C-ICs and their ability to promote tumour growth. A particularly novel contribution is the demonstration that the effect of an EZH2 inhibitor, UNC1999, is mediated to a significant extent through depression of the bivalently marked PcG target gene, Indian Hedgehog (IHH) and activation of the hedgehog pathway. The authors provide convincing evidence in support of these conclusions. This study provides important insight into the mechanisms by which EZH2 inhibition inhibits tumour initiation and proliferation. Due to its relevance to both basic tumour biology and use of epigenetic inhibitors in cancer treatment, this paper should be of interest to a wide range of investigators.

I have only one comment that the authors should address.

Following in vivo treatment of xenograft-containing mice with UNC1999, tumours were removed, disrupted and limiting dilutions of cells were transplanted into recipient mice. Based on a 4-fold reduction in tumour initiating frequency by the UNC1999 treated cells, the authors conclude that the EZH2 inhibition resulted in "CC-IC exhaustion". This appears to mean that they were induced to differentiate, losing their stem cell state, but this could be more explicitly stated. Secondly, were the transplanted cells tested for viability? If they were not, then an alternative explanation for the reduction in tumour initiation is simply that there were fewer viable cells in the tumours treated with UNC1999.

A minor correction. At the top of p. 8, the text refers to "Supplementary Fig.4C-D". This should state "Supplementary Fig.3C-D".

Reviewer #2 (Remarks to the Author):

The manuscript presents a comprehensive analysis revealing a regulatory role of the epigenetic modulator EZH2 on the hedgehog (HH) signaling pathway in colorectal cancer initiating cells (C-ICs). The authors show that demethylation of H3K27me3 by EZH2 in these cells induces IHH signaling, reduction of stem cell-like properties and increasing differentiation. Based on this observation, novel therapeutic approaches are suggested for CRC involving epigenetic modulation of the signaling pathway.

The strength of the manuscript is the use of primary organoid 3D cultures derived from patient tumors. This avoids observation of epigenetic changes established during long-term cultivation of cell lines and opens the view for stem-cell like cells. Importantly, spheroid cultures seem to be enriched for the C-ICs. This is nicely confirmed by limiting dilution assays, and an association of EZH2 expression data in TCGA mRNA with expression signatures from colon crypt and top cells. Can the stem-like crypt signature be found in the RNA seq data of the spheroids after UNC1999 treatment?

RNAseq and ChIP experiments show that the IHH promotor is located in a bivalent chromatin region characterized by the repressive H3K27me3 and the activating H3K4me3 histone mark. Inhibition, knockdown or overexpression of IHH is reducing the H3K27me3 mark and activating IHH signaling. The resolution of Fig 4a is not so good, so it is difficult to follow the identification the 20 bivalent region. The same is valid for suppl table 3 which presents also only the final result.

The authors describe the observed EZH2 effects by the presence of the histone marks of bivalent chromatin; information on H3K4me3 is however only given in the genome-wide analysis. The authors should validate also the presence of H3K4me3 in the IHH promotor as they did for H3K27me3 and they should also verify that H3K4me3 is not affected by EZH2 treatment.

The discussion of DNA methylation in bivalent chromatin regions in cancer cells raises the question on DNA methylation of IHH in the 3D cultures but also in LS174T in Fig 1c and 2g. Both data on DNA and H3K4me3 methylation might also be helpful for researchers who want to target bivalent chromatin by inhibiting enzymes specific for H3K4me3.

Minor aspects:

On page 6, the authors state that EZH1 is not affected. This is based on the use of inhibitors with high EZH2 specificity but not by direct measurement of EZH1 and should either be softened or EZH1 data should be shown.

The references to suppl. Figures 3 (missing?), 4 (page 7) and 5 (page 8) seem to be interchanged, please clarify this.

The manuscript is well written with a clear structure although the discussion is long and could be more focused to the results. The repeated statement "for the first time" might be overselling the story.

Reviewer #3 (Remarks to the Author):

This manuscript shows that a bivalent epigenetic mechanism is involved in the maintenance of a stem-like state of colorectal cancer-initiating cells.

The authors demonstrate the role of EZH2, a catalytic subunit of PRC2, in maintaining the cancer-initiating cell state through repression of the bivalent promoter of Indian Hedgehog. Inhibition of EZH2 has an effect on the proliferation and the sphere forming capacity of the cancer-initiating cells in vitro, reduces tumour growth in vivo and increases chemosensitivity to 5FU.

This is interesting work for understanding a new mechanism that regulates the stem-like state in CRC and bears potentially important implication in terms of new therapeutic treatment.

The results are interesting and convincing, the data are generally clear and logically presented, however few issues need to be addressed:

- 1) Do the Authors know the mutational status of the CRC samples grown as PDO? It would be interesting add this info in the text, if they are available.
- 2) Fig. 1: Authors should add photos of the PDO following treatment.
- 3) Fig. 2A: Is there any association between EZH2 expression and clinical data of samples used in the TMA?
- 4) Fig. S2A: The cell cycle should be analysed with an additional method (EdU incorporation).
- 5) Fig. S3C shows only the HE and H3K27me3 IHC on the xenografts following UNC1999 treatment, this is not sufficient. Authors should include expression of proliferation and apoptosis markers by IHC. They should also include expression of differentiation and stem cells markers by IHC or qPCR.

- 6) Proliferation and apoptosis markers should be also shown following 5FU treatment.
- 7) Please ensure figures are correctly referred in text.

Reviewer #4 (Remarks to the Author):

The work from Lima-Fernandes and collaborators focuses on understanding the role of PRC2 and the histones marks associated to this complex in colorectal cancer-initiating cells (CC-IC). This is a relevant issue in the field of cancer and cancer therapeutics and the data presented is mainly clear and experiments adequately performed. In general, the initial drug screen to identify CC-IC is correct and the confirmation that PRC2 is relevant to support cancer stemness is important. However, there are multiple technical and methodological issues that need to be carefully revised if the manuscript is further considered for publication in NatComms:

- 1) The study of EZH2 and H3K27me3 in 343 CRC patient samples and public data from the TCGA is extremely preliminary and has nothing to do with the heading of the section that is: Targeting EZH2 reduces CC-IC-specific growth and sphere forming capacity. Because it is the only piece of work that include patient-related data, authors should reconsider where to place the data and the type of analysis that could re-enforce the importance of their results: correlations with clinical data, survival curves, tumor subtypes, among others.
- 2) In addition, it is unclear where H3K27me3 is localized in the normal colonic tissue but it seems primarily found in the differentiated cell compartment where EZH2 is not expressed. Is this the case? Better or higher magnification images have to be included and distribution of EZH2 and the histone mark specified and discussed.
- 3) Most of the ex-vivo experiments are performed with the POP92 and POP66 patient-derived tumoroids but there is little information about the characteristics of the initial tumors and the tumoroids derived. Information about the tumor subtypes and the mutational status is essential for an adequate interpretation of the results.
- 4) In figure 2D, authors treat spheres with an EZH2 inhibitor and measure a TCF/LEF reporter activity as a measure of CC-IC activity, which is totally partial. Transcriptional and IF analysis of a panel of CC-IC markers has to be done (I would suggest Lgr5, EphB2, Ascl2, Lrig, CD44, CD133, ALDH1 among others). Also additional inhibitors of EZH activity are required.
- 5) Information about the protocol for seeding the tumoroids in the different experiments is lacking. It is essential to know whether they have been disaggregated at the single cells level and

then cultivated at specific concentrations. Also the recovery period before the treatments is important.

6) In general, the in vivo protocol that involves injection of disaggregated cells subcutaneously could be greatly improved by orthotopic implantation of tumoroid-derived cells in the cecum of the animals.

7) At the end of section 3, author state that “results show that EZH2 inhibition targets the CC-IC population and favors CC-IC exhaustion”. This assumption needs to be confirmed by serial passaging of treated tumoroids as it cannot be sustained by a 3-4 fold reduction in tumor initiating frequency.

8) Authors mention CDX2 as a differentiation marker of the intestine. Although it is true that CDX2 is essential to drive intestinal differentiation during embryonic development and controls the anterior-posterior axis of the developing gastrointestinal tissue, other differentiation markers would better reflect the terminal differentiation in the adult intestinal tissue (loss of ki67, increase in maturation markers such as muc2, LYZ1, CAII, neuroendocrine markers, etc)

9) The text related with figure 4A needs to be reformulated for clarity. For example, it is unclear what are the genes inside each group and whether the association between groups is statistically significantly or just distributed randomly. For example, the number of up-regulated genes after treatment is much higher than the number of down-regulated genes what necessarily alters the distribution of the histone marks in each group.

10) In the limiting dilution in vivo assays it is mentioned “CB.17 female scid/scid mice were injected with 3×10^5 CC-IC enriched cells”. Authors need to describe the method that was used to enrich tumors in CC-IC cells. Have they sorted Lgr5+/CD44+ epithelial cells? Have they used other stem cell markers?

Reviewer #1 (Remarks to the Author):

According to the cancer stem cell model, a subset of tumour cells, referred to as Cancer Stem Cells (C-ICs), have a capacity for self-renewal and tumour initiation. Polycomb-group (PcG) proteins contribute to the maintenance of the stem cell state by maintaining the transcriptional repression of differentiation-promoting genes. The promoters of these genes typically are bivalently marked with the antagonistic H3K4me3 and H3K27me3 epigenetic tags. EZH2, the catalytic component of the PcG PRC2 complex is responsible for depositing H3K27me3, which leads to recruitment of PRC1 and transcriptional repression. EZH2 inhibitors have been developed and currently are being tested in multiple cancer clinical trials. In this study, the authors investigate the role of PcG-mediated repression in maintaining the proliferative capacity of C-ICs and their ability to promote tumour growth. A particularly novel contribution is the demonstration that the effect of an EZH2 inhibitor, UNC1999, is mediated to a significant extent through depression of the bivalently marked PcG target gene, Indian Hedgehog (IHH) and activation of the hedgehog pathway. The authors provide convincing evidence in support of these conclusions. This study provides important insight into the mechanisms by which EZH2 inhibition inhibits tumour initiation and proliferation. Due to its relevance to both basic tumour biology and use of epigenetic inhibitors in cancer treatment, this paper should be of interest to a wide range of investigators.

I have only one comment that the authors should address.

Following *in vivo* treatment of xenograft-containing mice with UNC1999, tumours were removed, disrupted and limiting dilutions of cells were transplanted into recipient mice. Based on a 4-fold reduction in tumour initiating frequency by the UNC1999 treated cells, the authors conclude that the EZH2 inhibition resulted in “CC-IC exhaustion”. This appears to mean that they were induced to differentiate, losing their stem cell state, but this could be more explicitly stated.

We would like to thank Reviewer 1 for appreciating the importance of our study and its relevance to the scientific community. Indeed, we refer to the decreased tumour initiating frequency *in vivo* as “CC-IC exhaustion” which implies that the stem-like state is lost and the cells differentiate. Our revised manuscript now contains a second passage LDA performed using the primary LDA tumours that were processed and reinjected into another set of mice at limiting dilution. We observe on the second passage serial LDA a further reduction in CC-IC frequency to 8-fold in UNC1999 compared to vehicle. This result indicates that not only was the effect on CC-ICs sustained but there was further decrease in the CC-IC frequency (4-fold in 1st passage LDA, 8-fold in 2nd passage LDA), indicating that additional CC-ICs were undergoing exhaustion and failing to sustain the tumour over serial passage even 5 weeks after the last UNC1999 treatment (revised Figure 3j). Given the long timespan between UNC1999 treatment and the start of the secondary LDA, this demonstrates that the effect of EZH2 inhibition is not readily reversible and continues to result in CC-IC exhaustion long after treatment has ended, and supports the hypothesis of an irreversible, differentiated-like state (ie CC-IC exhaustion).

Revised Figure 3h-j: *In vivo* serial LDAs following UNC1999 treatment. (h) Schematic of the serial passage LDAs, performed after UNC1999-treated or vehicle tumours were harvested. (i) 1st passage LDA shows 4 fold decrease in self-renewal. (j) 2nd passage LDA *in vivo* shows 8-fold reduction in self-renewal.

Secondly, were the transplanted cells tested for viability? If they were not, then an alternative explanation for the reduction in tumour initiation is simply that there were fewer viable cells in the tumours treated with UNC1999.

The transplanted cells were indeed tested for viability prior to injection into the *in vivo* LDA. Tumours are first harvested from Vehicle and UNC1999-treated mice, digested using collagenase, washed, and viable cells counted using standard trypan blue exclusion. We have updated the manuscripts' methods to clarify this point. Our 2nd passage LDA *in vivo* clearly shows that the decrease in CC-IC frequency is not due to a reduction in fewer viable cells. The 2nd passage LDA was performed using the tumours from the 1st passage LDA, 52 days after the last UNC1999 dose. Given the long timespan between UNC1999 treatment and the start of the secondary LDA, this demonstrates that the effect of EZH2 inhibition is not solely due to having fewer viable cells in UNC1999-treated tumours, and that EZH2 inhibition continues to result in CC-IC exhaustion long after treatment has ended.

A minor correction. At the top of p. 8, the text refers to "Supplementary Fig.4C-D". This should state "Supplementary Fig.3C-D".

We fixed this error and have updated the manuscript accordingly.

Reviewer #2 (Remarks to the Author):

The manuscript presents a comprehensive analysis revealing a regulatory role of the epigenetic modulator EZH2 on the hedgehog (HH) signaling pathway in colorectal cancer initiating cells (C-ICs). The authors show that demethylation of H3K27me3 by EZH2 in these cells induces IHH signaling, reduction of stem cell-like properties and increasing differentiation. Based on this observation, novel therapeutic approaches are suggested for CRC involving epigenetic modulation of the signaling pathway.

The strength of the manuscript is the use of primary organoid 3D cultures derived from patient tumors. This avoids observation of epigenetic changes established during long-term cultivation

of cell lines and opens the view for stem-cell like cells. Importantly, spheroid cultures seem to be enriched for the C-ICs. This is nicely confirmed by limiting dilution assays, and an association of EZH2 expression data in TCGA mRNA with expression signatures from colon crypt and top cells. Can the stem-like crypt signature be found in the RNA seq data of the spheroids after UNC1999 treatment?

Reviewer 2 has a good point, regarding the stem-like crypt signature in treated cells. Using Gene Set Enrichment Analysis performed on the UNC1999-RNAseq, our revised manuscript now shows a decreased enrichment with the Colon Crypt signature (Figure 4b)

Revised Figure 4b - GSEA performed on the differentially regulated transcripts from the RNA-seq and the Colon Crypt signature shows inverse correlation (Family Wise Error rate (FWER) $p < 0.05$).

Consistent with our hypothesis, we also observe an enrichment for GO terms of genes involved in cell differentiation in the UNC1999-upregulated genes (Fig4d). We also observe a decrease in KLF4, NANOG, and OCT4 by RT-qPCR (Fig5d), which have been reported to be associated with stem-like properties in CRC (PMID 23418515, 19657699 and 23085761). Our revised manuscript now also shows a decrease in Wnt target genes Axin2, CD44 and EPHB2, which are well-established stem cell genes in CRC (Supplementary Figure S5g).

Revised Supplementary Figure S5g – RT-qPCR of Wnt target genes Axin2, CD44, and EPHB2 following treatment with UNC2400, UNC1999 or recombinant IHH

RNAseq and ChIP experiments show that the IHH promoter is located in a bivalent chromatin region characterized by the repressive H3K27me3 and the activating H3K4me3 histone mark. Inhibition, knockdown or overexpression of IHH is reducing the H3K27me3 mark and activating IHH signaling. The resolution of Fig 4a is not so good, so it is difficult to follow the identification the 20 bivalent region. The same is valid for suppl table 3 which presents also only the final result.

We apologize for the low resolution of Figure 4a and have made changes to the methods (page 26) in an effort to clarify how the promoters were identified as bivalently marked. “Promoter areas as defined by -2.5kb - +0.5kb from the TSS, were first identified. Each promoter was then assessed for H3K4me3 marked, H3K27me3 marked, or bivalently marked if H3K4me3 and H3K27me3 directly overlap within that region.” In an effort to clarify the identification of bivalent regions, we have included in this rebuttal a schematic for the peak calling around the promoter areas (-2.5kb, +0.5kb from the Transcription start site (TSS)). We also include additional ChIP-seq tracks for H3K27me3, H3K4me3 and bivalently marked promoters, which include peak calling information.

Reviewed Figure 4a : RNA-seq Heatmap of log₂ fold change for all genes significantly upregulated (top panel) or downregulated (bottom panel) following UNC1999 treatment in POP92. Lower bars underneath the heatmap indicate ChIP-seq assessment for the presence of H3K4me3 peaks (green), H3K27me3 peaks (pink) or regions where both marks directly overlap (Bivalent; purple) in promoter regions (2.5kb upstream of TSS, 0.5kb downstream of TSS) for all significantly differentially expressed genes (negative binomial, *fdr*-corrected *q* < 0.05).

Schematic representation of how Bivalency was defined in our manuscript. Bivalency was defined as the presence of directly overlapping peaks of K4me3 and K27me3 in the promoter area, 2.5kb upstream of TSS, 0.5kb downstream of TSS

Supplementary Screenshots of ChIP-seq tracks showing H3K4me3, H3K27me3 and bivalently marked promoters. ChIP tracks scale represent fold enrichment over background (0-20 for H3K4me3, 0-12 for H3K27me3). Promoter region (red bar, delimited by the dotted lines) is defined as +0.5kb to -2.5kb from the Transcription Start Site (TSS). Bivalent promoters were defined as the presence of directly overlapping H3K4me3 and

H3K27me3 called peaks. Promoters carrying both marks but not overlapping were defined as mixed.

The authors describe the observed EZH2 effects by the presence of the histone marks of bivalent chromatin; information on H3K4me3 is however only given in the genome-wide analysis. The authors should validate also the presence of H3K4me3 in the IHH promotor as they did for H3K27me3 and they should also verify that H3K4me3 is not affected by EZH2 treatment.

The reviewer raises a good point. We have performed the ChIP-qPCR for K4me3 following UNC1999 treatment, and observe the same level of the mark in untreated vs treated samples. This shows that H3K4me3 remains unaffected by the EZH2 inhibitor. Our revised manuscript includes this data (Supplementary Fig. 4i).

Revised Supplementary Figure S4i : ChIP-qPCR for for H3K4me3 following treatment with DMSO or UNC1999 (3uM).

The discussion of DNA methylation in bivalent chromatin regions in cancer cells raises the question on DNA methylation of IHH in the 3D cultures but also in LS174T in Fig 1c and 2g. Both data on DNA and H3K4me3 methylation might also be helpful for researchers who want to target bivalent chromatin by inhibiting enzymes specific for H3K4me3.

The Reviewer raises an interesting point regarding DNA methylation. For this study, our focus is to investigate whether bivalently marked promoters control lineage commitment in CC-ICs. With regards to IHH, we observe an increase in IHH mRNA following UNC1999 in 2 patient spheroids and LS174T spheres (Figure 5a), which indicates that the *IHH* gene is indeed under control of K27me3, which we confirmed by ChIP-seq as well as ChIP-PCR (Figures 4a, 4e, Supplementary Figure 4h and Supplementary Figure 5a). It is our opinion that investigating additional methylation marks is beyond the scope of this particular paper. However, we hypothesize that a subset of H3K27me3 marked genes that do not increase in expression following UNC1999 might still be repressed through other mechanisms, particularly DNA methylation. Several papers have reported the benefits of dual inhibition (EZH2 and DNMTs) in other models (PMID 29130642, 25477340...). We are currently investigating, as part of a separate new study, the cooperation of DNA methylation and H3K27me3 in self-renewal and chemo-resistance.

Minor aspects:

On page 6, the authors state that EZH1 is not affected. This is based on the use of inhibitors with high EZH2 specificity but not by direct measurement of EZH1 and should either be softened or EZH1 data should be shown.

We have updated the manuscript to incorporate the Reviewers' suggestion of softening the tone of the statement regarding EZH1:

“ Furthermore, genetic knockdown of EZH2 using two different shRNAs also reduced the growth of CC-IC enriched cultures (**Fig. 1e-f**), in line with the results obtained with the EZH2 inhibitors. Importantly, following EZH2 knockdown CC-IC enriched cultures showed no further reduction in the number of viable cells in the presence of UNC1999 (**Fig. 1g**). Taken together, this confirms that the growth inhibitory effect of UNC1999 is a consequence of EZH2 inhibition.”

The references to suppl. Figures 3 (missing?), 4 (page 7) and 5 (page 8) seem to be interchanged, please clarify this.

We apologize for the missing/wrong references to supplementary figures and have updated the manuscript accordingly.

The manuscript is well written with a clear structure although the discussion is long and could be more focused to the results. The repeated statement “for the first time” might be overselling the story.

We updated the manuscript to address this.

Reviewer #3 (Remarks to the Author):

This manuscript shows that a bivalent epigenetic mechanism is involved in the maintenance of a stem-like state of colorectal cancer-initiating cells.

The authors demonstrate the role of EZH2, a catalytic subunit of PRC2, in maintaining the cancer-initiating cell state through repression of the bivalent promoter of Indian Hedgehog. Inhibition of EZH2 has an effect on the proliferation and the sphere forming capacity of the cancer-initiating cells in vitro, reduces tumour growth in vivo and increases chemosensitivity to 5FU.

This is interesting work for understanding a new mechanism that regulates the stem-like state in CRC and bears potentially important implication in terms of new therapeutic treatment.

The results are interesting and convincing, the data are generally clear and logically presented, however few issues need to be addressed:

We would like to thank Reviewer 3 for the very positive comments and describing our study as interesting and convincing.

1) Do the Authors know the mutational status of the CRC samples grown as PDO? It would be interesting add this info in the text, if they are available.

We have included in the manuscript a summary of the mutational status for the PDOs used in the study, in Supplementary Table 2

	Origin	Stage	Mutations
LS174T	Primary Colon	II	KRAS, β -Catenin
POP92	Primary Colon	IV	APC, P53, BRAF
POP181	Lung Metastasis	IV	APC, P53, NRAS
POP66	Liver Metastasis	IV	BRAF, P53
POP164	Primary Colon	III	PIK3CA
CSC171C	Primary Colon	IV	APC, P53
CSC171L	Liver Metastasis	IV	APC, NRAS

Revised Supplementary Table2. List of all the cell models used in the study. The tissue of origin, stage of disease as well as major mutations are recapitulated

2) Fig. 1: Authors should add photos of the PDO following treatment.

We have included in the manuscript photos for the PDOs following treatment with UNC1999 after 7 days (Supplementary Figure S1c).

Revised Supplementary Figure S1c: Images of DMSO treated and UNC1999-treated organoids after 7 days.

3) Fig. 2A: Is there any association between EZH2 expression and clinical data of samples used in the TMA?

A couple of recent publications have reported correlation between EZH2 expression and clinical data in colorectal cancer. Ohuchi *et al* show that EZH2 expression increases during the progression of normal to adenoma and carcinoma, and positively correlates with Ki-67 (PMID

30214616). A recent publication by Carvalho *et al* also shows that EZH2 correlates with disease recurrence and progression (PMID 30105513). In line with these observations, we investigated the fraction of recurrence in the top and bottom quintiles of EZH2 expression. Our revised data now shows that EZH2 levels correlate with recurrence, with a larger proportion of EZH2-High patients showing recurrence (31%) vs EZH2-low patients (14%) (Fig. 2c). Altogether, our TMA points to the fact that EZH2 is indeed a relevant target in CRC and leads to the analysis in figure 2d-e, where we compare the levels of EZH2 transcript with colon top and colon stem signatures.

Revised Figure 2c: Percentage of recurrence within the top and bottom quintiles of EZH2 expression in the TMA. EZH2 High patients (IHC score >9.6) show a significantly higher percentage of recurrence (31.58%) compared to EZH2-low patients (IHC score <2.4) (14.46% recurrence). (p<0.01, Chi-square test).

4) Fig. S2a: The cell cycle should be analysed with an additional method (EdU incorporation).

We have added to our revised manuscript EdU incorporation + Hoechst dual staining for the validation of the cell cycle experiment. This new data supports our observation of decreased S phase (32-36 % decrease).

Supplementary Figure 2b : Cell cycle analysis using EdU incorporation and Hoechst

5) Fig. S3C shows only the HE and H3K27me3 IHC on the xenografts following UNC1999 treatment, this is not sufficient. Authors should include expression of proliferation and apoptosis markers by IHC. They should also include expression of differentiation and stem cells markers by IHC or qPCR.

We would like to thank the reviewer for suggesting to look at differentiation and Ki67 and assess proliferation in the UNC1999-treated xenografts. Indeed, tumours treated with UNC1999, had a 20% decrease in Ki67 staining (Supplementary figure 3d), which is on par with our observed fold-increase in CDX2 by IHC (Supplementary Figure 5c). Our manuscript also shows by qPCR the increase in CDX2 and FABP2, two differentiation markers.

With regards to apoptosis, our revised manuscript includes IHC for Cleaved Caspase 3, which shows increased staining in UNC1999 and combo groups (Supplementary figure 7c). We also have shown in Supplementary Figure 3c that UNC1999 treatment increases necrosis *in vivo*. The increase in necrosis suggests that more cells in the tumour have died following treatment.

Supplementary Figure S3d – Ki67 staining of Vehicle and UNC1999 treated tumours. Images show representative areas of the tumour. Scale bar is 200µm (40x). Quantification of Ki67-positive nuclei was performed on 5-6 randomly acquired fields per tumour, 4 tumours per group, and plotted on the right.

Supplementary Fig S5c-d – CDX2 differentiation marker staining in Vehicle and UNC1999-treated tumours

6) Proliferation and apoptosis markers should be also shown following 5FU treatment.

All of our cell viability measurements in spheroids are done through viable cell counting using Sytox Blue exclusion in Flow Cytometry. Below are the raw flow plots showing the decrease in viable cell count for 5-FU treatment alone and in combination with UNC1999.

Supplementary Figure : Raw Flow plots for viability assay in Figure 6a. Viable cells were counted using dead cell exclusion flow cytometry. Dead & dying cells are stained with Sytox Blue. Percentage live cells is written in the bottom right corner of the live cell gate.

Regarding the 5-FU treatments *in vivo*, our updated manuscript now shows H&E for all the treated groups of Figure 6, as well as cleaved Caspase 3. Our models are quite chemo-resistant, therefore the 5-FU treatment at 15mg/kg has no effect on tumour growth or self-renewal *in vivo*, as shown in Figure 6c-e. Therefore we shouldn't expect changes in apoptosis/proliferation, we also observe no changes in the levels of CDX2 and FABP2 (Supplementary Figure 7e-f). We observe a dramatic increase in necrosis and cleaved Caspase 3 in the combination group, as shown in the revised manuscript (Supplementary Figure 7c). The IHC results are supported by our finding that the xenografts treated with combined therapy (UNC1999 + 5-FU) demonstrated a significantly greater effect on tumour growth inhibition as compared to the individual treatments and control groups (Fig. 6c).

Supplementary Figure S7c : Cleaved caspase 3 staining assessed by IHC.

Supplementary Figure S7c : H&E of PDX from Figure 6

7) Please ensure figures are correctly referred in text.

We apologize for the wrong figure references in the text and have corrected the mistakes in the manuscript.

Reviewer #4 (Remarks to the Author):

The work from Lima-Fernandes and collaborators focuses on understanding the role of PRC2 and the histones marks associated to this complex in colorectal cancer-initiating cells (CC-IC). This is a relevant issue in the field of cancer and cancer therapeutics and the data presented is mainly clear and experiments adequately performed. In general, the initial drug screen to identify CC-IC is correct and the confirmation that PRC2 is relevant to support cancer stemness is important. However, there are multiple technical and methodological issues that need to be

carefully revised if the manuscript is further considered for publication in NatComms:

1) The study of EZH2 and H3K27me3 in 343 CRC patient samples and public data from the TCGA is extremely preliminary and has nothing to do with the heading of the section that is: Targeting EZH2 reduces CC-IC-specific growth and sphere forming capacity. Because it is the only piece of work that include patient-related data, authors should reconsider where to place the data and the type of analysis that could re-enforce the importance of their results: correlations with clinical data, survival curves, tumor subtypes, among others.

Our TMA points to the fact that EZH2 is indeed a relevant target in CRC and leads to the analysis in figure 2c, where we compare the levels of EZH2 transcript with colon top and colon stem signatures. The latter suggests that EZH2 levels are positively correlated with a large number of genes from the colon stem cell signature, and leads us to the next experiments in the manuscript to address the function of EZH2 in CC-ICs. Therefore we think that its placement in the manuscript is appropriate. In our revised manuscript we further re-enforce these results, by analyzing the fraction of patients whose cancer recurs in the top and bottom quintiles of EZH2 expression from our TMAs. We found that EZH2 levels correlate with recurrence, with a larger proportion of EZH2-High patients showing recurrence (31%) vs EZH2-low patients (14%). In further support of our findings several publications have reported correlation between EZH2 expression and clinical data in colorectal cancer. Ohuchi *et al* show that EZH2 expression increases during the progression of normal to adenoma and carcinoma, and positively correlates with Ki-67. A recent publication by Carvalho *et al* (PMID 30105513) also shows that EZH2 correlates with disease recurrence and progression, similar to our findings.

Revised Figure 2c : Percentage of recurrence within the top and bottom quintiles of EZH2 expression in the TMA. EZH2 High patients (IHC score >9.6) show a significantly higher percentage of recurrence (31.58%) compared to EZH2-low patients (IHC score <2.4) (14.46% recurrence).

2) In addition, it is unclear where H3K27me3 is localized in the normal colonic tissue but it seems primarily found in the differentiated cell compartment where EZH2 is not expressed. Is

this the case? Better or higher magnification images have to be included and distribution of EZH2 and the histone mark specified and discussed.

Unfortunately the immunohistochemical sections we obtained through the biobank were not all cut along the proper axis to be able to draw any conclusions with respect to H3K27me3 and EZH2 localization in the top and bottom part of the crypt. We agree with the reviewer that this is an interesting question but with the exception of the image shown in Figure 2a, none of our normal samples were cut along that axis. We unfortunately don't have samples cut on the right axis to make conclusions about the localization of EZH2 and H3K27me3.

3) Most of the ex-vivo experiments are performed with the POP92 and POP66 patient-derived tumoroids but there is little information about the characteristics of the initial tumors and the tumoroids derived. Information about the tumor subtypes and the mutational status is essential for an adequate interpretation of the results.

We have included in the manuscript a summary of the mutational status and tumour types for the PDOs used in the study. All of our PDO models were found to be sensitive to UNC1999, therefore we were not able to find correlations between tumour subtypes, mutational status and response to UNC1999.

4) In figure 2D, authors treat spheres with an EZH2 inhibitor and measure a TCF/LEF reporter activity as a measure of CC-IC activity, which is totally partial. Transcriptional and IF analysis of a panel of CC-IC markers has to be done (I would suggest Lgr5, EPhB2, Ascl2, Lrig, CD44, CD133, ALDH1 among others). Also additional inhibitors of EZH activity are required.

We agree with the reviewer's suggestion to look at additional Wnt target genes to confirm the TCF/LEF reporter activity. We observe a decrease in Axin2, CD44, and EPHB2 following UNC1999 treatment as well as recombinant IHH (Supplementary Figure 5g).

Supplementary Figure S5g – RT-qPCR of Wnt target genes Axin2, CD44, and EPHB2 following treatment with UNC2400, UNC1999 or recombinant IHH

With regards to the decrease in TCF/LEF reporter activity being partial, we would like to note that the maximum decrease we have ever observed previously using this GFP Wnt reporter with a known WNT pathway inhibitor such as iCRT14, is around 55% decrease in reporter activity (Kreso et al, Nat Med 2015, PMID 24292392). We have also included GSK343 which shows a drop in reporter activity by up to 50% following treatment (Supplementary Figure 2e-f).

Supplementary Figure S2 d-g : TCF/LEF Wnt reporter now shows decrease in activity using GSK343 (3rd panel in (d), updated table in (e), and Normalized fold change in (f).

5) Information about the protocol for seeding the tumoroids in the different experiments is lacking. It is essential to know whether they have been disaggregated at the single cells level and then cultivated at specific concentrations. Also the recovery period before the treatments is important.

We agree with the Reviewer, it is important to specify the technical protocols with PDOs. We have included in the manuscript a more detailed version of the protocol for the PDO experiments. Briefly, PDOs were dissociated down to the single cell level and seeded at a density of 1000 cells per well in a 384 well plate, coated with matrigel. The cells were left to recover for 24h before treatment was added. Plates were read out using Celltiterglo 7 days following treatment. (Methods, “Epigenetic probes screen and Viable cell count assays, page 20).

6) In general, the in vivo protocol that involves injection of disaggregated cells subcutaneously could be greatly improved by orthotopic implantation of tumoroid-derived cells in the cecum of the animals.

We completely agree with the reviewer that ideally cells should also be injected orthotopically when possible. We are working on developing a reproducible orthotopic model for colorectal cancer which involves injecting the cells into the cecal wall. However, to date, the model has not been reliable which makes large scale tumour growth inhibition experiments impossible. It is for this reason we have chosen to do these experiments using a subcutaneous

injection model. There are very few publications that utilize an orthotopic CRC murine xenograft model to do preclinical drug testing, which speaks to how challenging this model can be.

7) At the end of section 3, author state that “results show that EZH2 inhibition targets the CC-IC population and favors CC-IC exhaustion”. This assumption needs to be confirmed by serial passaging of treated tumouroids as it cannot be sustained by a 3-4 fold reduction in tumor initiating frequency.

The Reviewer raises an excellent point with regards to assessing CC-IC exhaustion. While the manuscript was under peer-review to Nature Communications, we had a 2nd passage LDA in progress, which was performed on cells derived from the primary passage LDA (Figure 3). Interestingly, the secondary passage LDA demonstrated a greater reduction in the CC-IC numbers as compared to the primary LDA (8-fold reduction in 2nd passage LDA versus 4-fold reduction in 1st passage LDA) indicating that increasing numbers of CC-ICs continue to undergo exhaustion even 5 weeks after treatment with UNC1999 ended (revised Figure 3j). Given the long timespan between UNC1999 treatment and the start of the secondary LDA, this demonstrates that the effect of EZH2 inhibition is not readily reversible and continues to result in CC-IC exhaustion long after treatment has ended.

Revised Figure 3h-j: *In vivo* serial LDAs following UNC1999 treatment. (h) Schematic of the serial passage LDAs, performed after UNC1999-treated or vehicle tumours were harvested. (i) 1st passage LDA shows 4 fold decrease in self-renewal. (j) 2nd passage LDA *in vivo* shows 8-fold reduction in self-renewal.

8) Authors mention CDX2 as a differentiation marker of the intestine. Although it is true that CDX2 is essential to drive intestinal differentiation during embryonic development and controls the anterior-posterior axis of the developing gastrointestinal tissue, other differentiation markers would better reflect the terminal differentiation in the adult intestinal tissue (loss of ki67, increase in maturation markers such as muc2, LYZ1, CAII, neuroendocrine markers, etc)

The reviewer raises an interesting point to look at additional differentiation markers for the different subtypes of differentiated cells in the colon. Our manuscript initially showed an

increase for FABP2 and CDX2 by mRNA, two markers of differentiation for the enterocytic lineage. To address the reviewer's comment, we now performed an additional RT-qPCR to determine whether the CC-ICs differentiate toward the Goblet or Enteroendocrine lineages and show that no significant changes are detected following UNC1999 or recombinant IHH treatment for MUC2 (Goblet) or REG4 (Enteroendocrine) (See panel below). Our results indicate that CC-ICs exit the stem-like state to a differentiated-like state resembling enterocytes (Fig. 5c, Supplementary Figure 5c). We have not included LYZ1 marker as it is a marker for Paneth Cells which are not present in the colon crypt. We confirmed our CDX2 qPCR by performing IHC for CDX2 in vehicle and UNC1999-treated xenograft tumours, showing a significant increase of CDX2 at the protein level following UNC1999 treatment (Supplementary Figure 5c).

Supplementary figure : RT-qPCR for Goblet cell marker Muc2 and Enteroendocrine marker Reg4 following treatment with UNC2400, UNC1999, and recombinant IHH.

Supplementary Fig S5c-d– CDX2 differentiation marker staining in Vehicle and UNC1999-treated tumours

Several reports have shown that CDX2 is a differentiation marker in CRC, including a recent publication by Dalerba et al, 2016 in New England Journal of Medicine (<http://www.nejm.org/doi/full/10.1056/NEJMoa1506597>). In this article the authors used a bioinformatics approach to search for biomarkers of colon epithelial differentiation across gene-expression arrays, and the top biomarker of colon epithelial differentiation that they identified was CDX2. Lack of CDX2 expression in CRC identified a subset of high risk stage II CRCs that were poorly differentiated and associated with a decreased disease specific survival.

Our increase in CDX2 expression is now supported in the manuscript by a decrease in Ki67 staining in UNC1999-treated tumours (Supplementary Figure 3d), suggesting a decrease in proliferative capacity. Moreover, we observe a decrease in KLF4, NANOG, and OCT4, which have been reported to be associated with stem-like properties in CRC (PMID 23418515, 19657699 and 23085761). Our revised manuscript now also shows a decrease in Wnt target genes Axin2, CD44 and EPHB2, which are well-established stem cell genes in CRC (Supplementary Figure 5g) .

Supplementary Figure S3d – Ki67 staining in Vehicle and UNC1999-treated tumours

Supplementary Figure S5g – RT-qPCR of Wnt target genes Axin2, CD44, and EPHB2 following treatment with UNC2400, UNC1999 or recombinant IHH

9) The text related with figure 4A needs to be reformulated for clarity. For example, it is unclear what are the genes inside each group and whether the association between groups is statistically significant or just distributed randomly. For example, the number of up-regulated genes after treatment is much higher than the number of down-regulated genes what necessarily alters the distribution of the histone marks in each group.

We have updated our annotation of Fig 4a and revised the manuscript accordingly. It is important to keep in mind that in this particular study, we are inhibiting EZH2, a repressor of transcription. It is therefore not surprising that we observe a larger set of genes upregulated after UNC1999 treatment (the result of decreased transcriptional repression), compared to the number of downregulated genes. The latter are most likely secondary/downstream consequences of EZH2 inhibition; for example a decrease in expression of proliferation genes can be the result of the de-repression of a differentiation program.

The histone marks K27me3 and K4me3 were assessed in Figure 4a in order to help identify the initial chromatin state of EZH2 targets in our model. We showed in figures 4a and 4b that transcriptional upregulation following UNC1999 is highly likely in promoters carrying H3K27me3 (particularly in combination with H3k4me3), This indicates that K27me3 as well as bivalent marks are predictive of a specific gene being upregulated following UNC1999.

The reviewer raises a good question with regards to whether the association between each group is statistically significant or distributed randomly, our supplementary Figure 4f addresses this. Indeed, by performing a gene set enrichment analysis (GSEA), we show that the likelihood of a gene to be upregulated following UNC1999 treatment is very high in the Bivalent and H3K27me3 marked promoters. There is no correlation between the status of H3K4me3 and UNC1999 up-or downregulation. We have updated the manuscript to clarify this point.

10) In the limiting dilution *in vivo* assays it is mentioned “CB.17 female scid/scid mice were injected with 3x10⁵ CC-IC enriched cells”. Authors need to describe the method that was used to enrich tumors in CC-IC cells. Have they sorted Lgr5+/CD44+ epithelial cells? Have they used other stem cell markers?

The CC-IC enriched models used in this study were described in previous studies by our group, published in Cancer Cell (PMID 22698403) and Nature Medicine (PMID 24292392). Enrichment of CC-ICs was confirmed phenotypically using *in vivo* limited dilution assays (LDAs). We have updated the manuscript to include more details of these methods.

Briefly, CC-IC enriched spheroids were generated from CRC patient-derived xenografts: PDX models were made using tumours obtained from patients at the time of surgical resection.. PDX models were then dissociated, depleted for mouse cells, and cultured in suspension using serum-free growth factor enriched media as spheroids (Anoikis-resistant cultures). In parallel, some cells were cultured in 10% serum-containing media yielding adherent cells which are not enriched for CC-ICs. When the spheroid model was established, we confirmed that they were CC-IC enriched, by performing the gold-standard *in vivo* Limiting Dilution Assay. We injected

the cells at limiting dilution in mice and showed that the cells grown as spheres in serum-free, growth factor enriched media, were enriched in tumour initiating cells, as compared to the same cells grown in 10% serum (See table below, extracted from O'Brien et al, Cancer Cell (PMID 22698403)).

Table S5: CC-ICs are enriched in cultures grown in sphere versus colony forming media, related to Figure 1.

Colon cancer sample	Culture condition	No. of cells injected	No. of injections	No. of tumors formed	CC-IC frequency 1 in x		
					Lower	Estimate	Upper
LS174T	with serum	500,000	12	11	31,278	15,166	7,354
		100,000	12	11			
		10,000	12	9			
		1,000	12	12			
		100	12	8			
49367G1P2	with serum	1,000,000	12	12	6,873	3,527	1,810
		100,000	12	12			
		10,000	12	8			
		1,000	12	8			
		100	12	6			
49367G1P2	serum free	100,000	12	12	113	56	28
		10,000	12	12			
		1,000	12	12			
		100	12	10			

Table S5, extracted from O'Brien et al, Cancer Cell (PMID 22698403), showing decrease in CC-IC frequency in serum cultures vs serum free spheroids.

REVIEWERS' COMMENTS:

Reviewer #1 (Remarks to the Author):

The authors have satisfactorily addressed my comments. In my opinion, the manuscript is now suitable for publication in Nature Communications.

Reviewer #2 (Remarks to the Author):

The manuscript provides strong data to suggest novel approaches for CRC treatment including epigenetic modeling of signaling pathways. The concerns of this reviewer have been addressed adequately and the paper should be published in its current form.

Reviewer #3 (Remarks to the Author):

The authors have thoroughly addressed my previous comments and I now recommend the manuscript is accepted for publication.

Reviewer #4 (Remarks to the Author):

Authors have properly addressed my initial concerns. I find that the current version is greatly improved and suitable for publication in NatComms.